# Neural Active Learning with Performance Guarantees

**Zhilei Wang**
New York University
New York, NY 10012
zhileiwang92@gmail.com

**Pranjal Awasthi**
Google Research
New York, NY 10011
pranjalawasthi@google.com

**Christoph Dann**
Google Research
New York, NY 10011
chrisdann@google.com

**Ayush Sekhari**
Cornell University
Ithaca, NY 14850
ayush.sekhari@gmail.com

**Claudio Gentile**
Google Research
New York, NY 10011
cgentile@google.com

## Abstract

We investigate the problem of active learning in the streaming setting in non-parametric regimes, where the labels are stochastically generated from a class of functions on which we make no assumptions whatsoever. We rely on recently proposed Neural Tangent Kernel (NTK) approximation tools to construct a suitable neural embedding that determines the feature space the algorithm operates on and the learned model computed atop. Since the shape of the label requesting threshold is tightly related to the complexity of the function to be learned, which is a-priori unknown, we also derive a version of the algorithm which is agnostic to any prior knowledge. This algorithm relies on a regret balancing scheme to solve the resulting online model selection problem, and is computationally efficient. We prove joint guarantees on the cumulative regret and number of requested labels which depend on the complexity of the labeling function at hand. In the linear case, these guarantees recover known minimax results of the generalization error as a function of the label complexity in a standard statistical learning setting.

## 1 Introduction

Supervised learning is a fundamental paradigm in machine learning and is at the core of modern breakthroughs in deep learning [29]. A machine learning system trained via supervised learning requires access to labeled data collected via recruiting human experts, crowdsourcing, or running expensive experiments. Furthermore, as the complexity of current deep learning architectures grows, their requirement for labeled data increases significantly. The area of *active learning* aims to reduce this data requirement by studying the design of algorithms that can learn and generalize from a small carefully chosen subset of the training data [13, 40].

The two common formulations of active learning are *pool based* active learning, and *sequential (or streaming)* active learning. In the pool based setting [30], the learning algorithm has access to a large unlabeled set of data points, and the algorithm can ask for a subset of the data to be labeled. In contrast, in the sequential setting, data points arrive in a streaming manner, either adversarially or drawn i.i.d. from a distribution, and the algorithm must decide whether to query the label of a given point or not [14].

From a theoretical perspective, active learning has typically been studied under models inspired by the probably approximately correct (PAC) model of learning [41]. Here one assumes that there is a pre-specified class $\mathcal{H}$ of functions such that the *target* function mapping examples to their labels

either lies in $\mathcal{H}$ or has a good approximation inside the class. Given access to unlabeled samples generated i.i.d. from the distribution, the goal is to query for a small number of labels and produce a hypothesis of low error.

In the *parametric* setting, namely, when the class of functions $\mathcal{H}$ has finite VC-dimension (or finite disagreement coefficient) [21], the rate of convergence of active learning, i.e., the rate of decay of the regret as a function of the number of label queries ($N$), is of the form $\nu N^{-1/2} + e^{-\sqrt{N}}$, where $\nu$ is the population loss of the best function in class $\mathcal{H}$. This simple finding shows that active learning behaves like passive learning when $\nu > 0$, while very fast rates can only be achieved under low noise ($\nu \approx 0$) conditions. This has been worked out in, e.g., [19, 15, 5, 4, 6, 38].

While the parametric setting comes with methodological advantages, the above shows that in order to unleash the true power of active learning, two properties are desirable: (1) A better interplay between the input distribution and the label noise and, (2) a departure from the parametric setting leading us to consider wider classes of functions (so as to reduce the population loss $\nu$ to close to 0). To address the above, there has also been considerable theoretical work in recent years on non-parametric active learning [10, 33, 31]. However, these approaches suffer from the curse of dimensionality and do not lead to computationally efficient algorithms. A popular approach that has been explored empirically in recent works is to use Deep Neural Networks (DNNs) to perform active learning (e.g., [37, 26, 39, 3, 44]). While these works empirically demonstrate the power of the DNN-based approach to active learning, they do not come with provable guarantees. The above discussion raises the following question: *Is provable and computationally efficient active learning possible in non-parametric settings?*

We answer the above question in the affirmative by providing the first, to the best of our knowledge, computationally efficient algorithm for active learning based on Deep Neural Networks. Similar to non-parametric active learning, we avoid fixing a function class a-priori. However, in order to achieve computational efficiency, we instead propose to use over-parameterized DNNs, where the amount of over-parameterization depends on the input data at hand. We work in the sequential setting, and propose a simple active learning algorithm that forms an uncertainty estimate for the current data point based on the output of a DNN, followed by a gradient descent step to update the network parameters if the data point is queried. We show that under standard low-noise assumptions [32] our proposed algorithm achieves fast rates of convergence.

In order to analyze our algorithm, we use tools from the theory of Neural Tangent Kernel (NTK) approximation [24, 2, 18] that allows us to analyze the dynamics of gradient descent by considering a linearization of the network around random initialization. Since we study the non-parametric regime, the convergence rates of our algorithm depend on a data-dependent complexity term that is expected to be small in practical settings, but could be very large in worst-case scenarios. Furthermore, the algorithm itself needs an estimate of complexity term in order to form accurate uncertainty estimates. We show that one can automatically adapt to the magnitude of the unknown complexity term by designing a novel model selection algorithm inspired by recent works in model selection in multi-armed bandit settings [36, 35]. Yet, several new insights are needed to ensure that the model selection algorithm can simultaneously achieve low generalization error without spending a significant amount of budget on label queries.

## 2   Preliminaries and Notation

Let $\mathcal{X}$ denote the input space, $\mathcal{Y}$ the output space, and $\mathcal{D}$ an unknown distribution over $\mathcal{X} \times \mathcal{Y}$. We denote the corresponding random variables by $x$ and $y$. We also denote by $\mathcal{D}_{\mathcal{X}}$ the marginal distribution of $\mathcal{D}$ over $\mathcal{X}$, and by $\mathcal{D}_{\mathcal{Y}|x_0}$ the conditional distribution of random variable $y$ given $x = x_0$. Moreover, given a function $f$ (sometimes called a hypothesis or a model) mapping $\mathcal{X}$ to $\mathcal{Y}$, the conditional *population loss* (often referred to as conditional *risk*) of $f$ is denoted by $L(f \mid x)$, and defined as $L(f \mid x) = \mathbb{E}_{y \sim \mathcal{D}_{\mathcal{Y}|x}}[\ell(f(x), y) \mid x]$, where $\ell : \mathcal{Y} \times \mathcal{Y} \to [0, 1]$ is a *loss* function. For ease of presentation, we restrict to a binary classification setting with 0-1 loss, whence $\mathcal{Y} = \{-1, +1\}$, and $\ell(a, y) = \mathbb{1}\{a \neq y\} \in \{0, 1\}$, $\mathbb{1}\{\cdot\}$ being the indicator function of the predicate at argument. When clear from the surrounding context, we will omit subscripts like "$y \sim \mathcal{D}_{\mathcal{Y}|x}$" from probabilities and expectations.

We investigate a *non-parametric* setting of active learning where the conditional distribution of $y$ given $x$ is defined through an unknown function $h : \mathcal{X}^2 \to [0,1]$ such that

$$\mathbb{P}(y = 1 \,|\, x) = h((x,0)) \qquad \mathbb{P}(y = -1 \,|\, x) = h((0,x)) \,, \tag{1}$$

where $0 \in \mathcal{X}$, $(x_1, x_2)$ denotes the concatenation (or pairing) of the two instances $x_1$ and $x_2$ (so that $(x,0)$ and $(0,x)$ are in $\mathcal{X}^2$) and, for all $x \in \mathcal{X}$ we have $h((x,0)) + h((0,x)) = 1$. We make no explicit assumptions on $h$, other than its well-behavedness w.r.t. the data $\{x_t\}_{t=1}^T$ at hand through the formalism of Neural Tangent Kernels (NTK) – see below. As a simple example, in the linear case, $\mathcal{X}$ is the $d$-dimensional unit ball, $h(\cdot, \cdot)$ is parametrized by an unknown unit vector $\theta \in \mathbb{R}^d$, and $h((x_1, x_2)) = \frac{1 + \langle (\theta, -\theta), (x_1, x_2) \rangle}{2}$ , so that $h((x,0)) = \frac{1 + \langle \theta, x \rangle}{2}$ and $h((0,x)) = \frac{1 - \langle \theta, x \rangle}{2}$, where $\langle \cdot, \cdot \rangle$ is the usual dot product in $\mathbb{R}^d$.

We consider a streaming setting of active learning where, at each round $t \in [T] = \{1, \ldots, T\}$, a pair $(x_t, y_t) \in \mathcal{X} \times \mathcal{Y}$ is drawn i.i.d. from $\mathcal{D}$. The learning algorithm receives as input only $x_t$, and is compelled to both issue a prediction $a_t$ for $y_t$ and, at the same time, decide on-the-fly whether or not to observe $y_t$. These decisions can only be based on past observations. Let $\mathbb{E}_t$ denote the conditional expectation $\mathbb{E}[\cdot \,|\, (x_1, y_1) \ldots, (x_{t-1}, y_{t-1}), x_t]$, and we introduce the shorthand

$$x_{t,a} = \begin{cases} (x_t, 0) & \text{if } a = 1 \\ (0, x_t) & \text{if } a = -1 \,. \end{cases}$$

Notice that with this notation $\mathbb{E}[\ell(a, y_t) \,|\, x_t] = 1 - h(x_{t,a})$, for all $a \in \mathcal{Y}$. We quantify the accuracy of the learner's predictions through its (pseudo) *regret*, defined as

$$R_T \;=\; \sum_{t=1}^T \Big( \mathbb{E}_t[\ell(a_t, y_t) \,|\, x_t] - \mathbb{E}[\ell(a_t^*, y_t) \,|\, x_t] \Big) \;=\; \sum_{t=1}^T \big( h(x_{t,a_t^*}) - h(x_{t,a_t}) \big) \,,$$

where $a_t^*$ is the Bayesian-optimal classifier on instance $x_t$, that is, $a_t^* = \arg\max_{a \in \mathcal{Y}} h(x_{t,a})$. Additionally, we are interested in bounding the number of labels $N_T$ the algorithm decides to request. Our goal is to *simultaneously* bound $R_T$ and $N_T$ with high probability over the generation of the sample $\{(x_t, y_t)\}_{t=1,\ldots,T}$ .

Throughout this work, we consider the following common low-noise condition on the marginal distribution $\mathcal{D}_\mathcal{X}$ (Mammen-Tsybakov low noise condition [32]): There exist absolute constants $c > 0$, and $\alpha \geq 0$ such that for all $\epsilon \in (0, 1/2)$ we have

$$\mathbb{P}\big( |h((x,0)) - \tfrac{1}{2}| < \epsilon \big) \leq c\,\epsilon^\alpha \,.$$

In particular, $\alpha = \infty$ gives the so-called *hard margin* condition $\mathbb{P}\big( |h((x,0)) - \tfrac{1}{2}| < \epsilon \big) = 0$. while, at the opposite extreme, exponent $\alpha = 0$ (and $c = 1$) results in *no assumptions whatsoever* on $\mathcal{D}_\mathcal{X}$. For simplicity, we shall assume throughout that the above low-noise condition holds for[1] $c = 1$.

Our techniques are inspired by the recent work [45] from which we also borrow some notation. We are learning the class of functions $\{h\}$ by means of fully connected neural networks

$$f(x, \theta) = \sqrt{m} W_n \sigma(\ldots \sigma(W_1 x)) \,,$$

where $\sigma$ is a ReLU activation function $\sigma(x) = \max\{0, x\}$, $m$ is the width of the network and $n \geq 2$ is its depth. In the above, $\theta \in \mathbb{R}^p$ collectively denotes the set of weights $\{W_1, W_2, \ldots, W_n\}$ of the network, where $p = m + 2md + m^2(n-2)$ is their number, and the input $x$ at training time should be thought of as some $x_{t,a} \in \mathcal{X}^2$.

With any depth-$n$ network and data points $\{x_{t,a}\}_{t=1,\ldots,T,\,a=\pm 1}$ we associate a depth-$n$ NTK matrix as follows [24]. First, rename $\{x_{t,a}\}_{t=1,\ldots,T,\,a=\pm 1}$ as $\{x^{(i)}\}_{i=1,\ldots,2T}$. Then define matrices

$$\widetilde{H}^{(1)} = \Big[ H_{i,j}^{(1)} \Big]_{i,j=1}^{2T \times 2T} \qquad \Sigma^{(1)} = \Big[ \Sigma_{i,j}^{(1)} \Big]_{i,j=1}^{2T \times 2T} \qquad \text{with} \qquad H_{i,j}^{(1)} = \Sigma_{i,j}^{(1)} = \langle x^{(i)}, x^{(j)} \rangle \,,$$

and then, for any $k \leq n$ and $i, j = 1, \ldots, 2T$, introduce the bivariate covariance matrix

$$A_{i,j}^{(k)} = \begin{bmatrix} \Sigma_{i,i}^{(k)} & \Sigma_{i,j}^{(k)} \\ \Sigma_{i,j}^{(k)} & \Sigma_{j,j}^{(k)} \end{bmatrix}$$

---

[1]A more general formulation requires the above to hold only for $\epsilon \leq \epsilon_0$, where $\epsilon_0 \in (0, 1/2)$ is a third parameter. We shall omit this extra parameter from our presentation.

by which we recursively define

$$\Sigma_{i,j}^{(k+1)} = 2\mathbb{E}_{(u,v)\sim N(0,A_{i,j}^{(k)})}[\sigma(u)\sigma(v)]$$

and

$$\widetilde{H}_{i,j}^{(k+1)} = 2\widetilde{H}_{i,j}^{(k)}\mathbb{E}_{(u,v)\sim N(0,A_{i,j}^{(k)})}[\,\mathbb{1}\{u \geq 0\}\,\mathbb{1}\{v \geq 0\}] + \Sigma_{i,j}^{(k+1)}\,.$$

The $2T \times 2T$-dimensional matrix $H = \frac{1}{2}(\widetilde{H}^{(n)} + \Sigma^{(n)})$ is called the Neural Tangent Kernel (NTK) matrix of depth $n$ (and infinite width) over the set of points $\{x_{t,a}\}_{t=1,\ldots,T,\,a=\pm1}$. The reader is referred to [24] for more details on NTK.

In order to avoid heavy notation, we assume $||x_t|| = 1$ for all $t$. Matrix $H$ is positive semi-definite by construction but, as is customary in the NTK literature (e.g., [2, 9, 17]), we assume it is actually positive definite (hence invertible) with smallest eigenvalue $\lambda_0 > 0$. This is a mild assumption that can be shown to hold if no two vectors $x_t$ are aligned to each other.

We measure the complexity of the function $h$ at hand in a way similar to [45]. Using the same rearrangement of $\{x_{t,a}\}_{t=1,\ldots,T,\,a=\pm1}$ into $\{x^{(i)}\}_{i=1,\ldots,2T}$ as above, let $\mathbf{h}$ be the $2T$-dimensional (column) vector whose $i$-th component is $h(x^{(i)})$. Then, we define the complexity $S_{T,n}(h)$ of $h$ over $\{x_{t,a}\}_{t=1,\ldots,T,\,a=\pm1}$ w.r.t. an NTK of depth $n$ as $S_{T,n}(h) = \sqrt{\mathbf{h}^\top H^{-1}\mathbf{h}}$. Notice that this notion of (data-dependent) complexity is consistent with the theoretical findings of [2], who showed that for a two-layer network the bound on the generalization performance is dominated by $\mathbf{y}^\top H^{-1}\mathbf{y}$, where $\mathbf{y}$ is the vector of labels. Hence if $\mathbf{y}$ is aligned with the top eigenvectors of $H$ the learning problem becomes easier. In our case, vector $\mathbf{h}$ plays the role of vector $\mathbf{y}$. Also observe that $S_{T,n}^2(h)$ can in general be as big as linear in $T$ (in which case learning becomes hopeless with our machinery). In the special case where $h$ belongs to the RKHS induced by the NTK, one can upper bound $S_{T,n}(h)$ by the norm of $h$ in the RKHS.

The complexity term $S_{T,n}(h)$ is typically *unknown* to the learning algorithm, and it plays a central role in both regret and label complexity guarantees. Hence the algorithm needs to *learn* this value as well during its online functioning. Apparently, this aspect of the problem has been completely overlooked by [45] (as well as by earlier references on contextual bandits in RKHS, like [12]), where a (tight) upper bound on $S_{T,n}(h)$ is assumed to be available in advance. We will cast the above as a *model selection* problem in active learning, where we adapt and largely generalize to active learning the regret balancing technique from [36, 35].

In what follows, we use the short-hand $g(x;\theta) = \nabla_\theta f(x,\theta)$ and, for a vector $g \in \mathbb{R}^p$ and matrix $Z \in \mathbb{R}^{p\times p}$, we often write $\sqrt{g^\top Z g}$ as $||g||_Z$, so that $S_{T,n}(h) = ||\mathbf{h}||_{H^{-1}}$.

## 2.1 Related work

The main effort in theoretical works in active learning is to obtain rates of convergence of the population loss of the hypothesis returned by the algorithm as a function of the number $N$ of requested labels. We emphasize that most of these works, that heavily rely on approximation theory, are *not* readily comparable to ours, since our goal here is not to approximate $h$ through a DNN on the entire input domain, but only on the data at hand.

As we recalled in the introduction, in the *parametric* setting the convergence rates of the regret are of the form $\nu\,N^{-1/2} + e^{-\sqrt{N}}$, where $\nu$ is the population loss of the best function in class $\mathcal{H}$. Hence, active learning rates behave like the passive learning rate $N^{-1/2}$ when $\nu > 0$, while fast rates can only be achieved under very low noise ($\nu \approx 0$) conditions. In this respect, relevant references include [20, 27] where, e.g., in the realizable case (i.e., when the Bayes optimal classifier lies in $\mathcal{H}$), minimax active learning rates of the form $N^{-\frac{\alpha+1}{2}}$ are shown to hold for adaptive algorithms that do not know beforehand the noise exponent $\alpha$. In non-parametric settings, a comprehensive set of results has been obtained by [31], which builds on and significantly improves over earlier results from [33]. Both papers work under smoothness (Holder continuity/smoothness) assumptions. In addition, [33] requires $\mathcal{D}_\mathcal{X}$ to be (quasi-)uniform on $\mathcal{X} = [0,1]^d$. In [31] the minimax active learning rate $N^{-\frac{\beta(\alpha+1)}{2\beta+d}}$ is shown to hold for $\beta$-Holder classes, where exponent $\beta$ plays the role of the complexity of the class of functions to learn, and $d$ is the input dimension. This algorithm is adaptive to the complexity parameter $\beta$, and is therefore performing a kind of model selection. Notice that minimax rates in the

parametric regime are recovered by setting $\beta \to \infty$. Of a somewhat similar flavor is an earlier result by [27], where a convergence rate of the form $N^{-\frac{\alpha+1}{2+\kappa\alpha}}$ is shown, being $\kappa$ the metric entropy of the class (again, a notion of complexity). A refinement of the results in [31] has recently been obtained by [34] where, following [11], a more refined notion of smoothness for the Bayes classifier is adopted which, however, also implies more restrictive assumptions on the marginal distribution $\mathcal{D}_\mathcal{X}$.

As opposed to those bounds, our bounds are *data-dependent*, in that all relevant quantities appearing in the bounds will be random variables depending on the data at hand (which are themselves random). One may attempt to turn these into *data-independent* results (like in most of the papers we cited above) by, e.g., establishing bounds on that hold in expectation or with high probability over the random draw of the data, but this theory is currently unavailable in the NTK literature (as far as we know). Very recently some results have appeared for certain special cases, see [23] for example. But such results are too embryonic in nature to allow us a full-fledged comparison.

Model selection of the scale of a Nearest-Neighbor-based active learning algorithm is also performed in [28], whose main goal is to achieve data-dependent rates based on the noisy-margin properties of the random sample at hand, rather than those of the marginal distribution. Their active learning rates are not directly comparable to ours and, unlike our paper, the authors work in a *pool-based* scenario, where all unlabeled points are available beforehand. Finally, an interesting investigation in active learning for over-parametrized and interpolating regimes is contained in [25]. The paper collects a number of interesting insights in active learning for 2-layer Neural Networks and Kernel methods, but it restricts to either uniform distributions on the input space or cases of well-clustered data points, with no specific regret and query complexity guarantees, apart from very special (though insightful) cases.

## 3   Basic Algorithm

Our first algorithm (Algorithm 1) uses randomly initialized, but otherwise frozen, network weights (a more refined algorithm where the network weights are updated incrementally is described and analyzed in the appendix). Algorithm 1 is an adaptation to active learning of the neural contextual bandit algorithm of [45], and shares similarities with an earlier selective sampling algorithm analyzed in [16] for the linear case. The algorithm generates network weights $\theta_0$ by independently sampling from Gaussian distributions of appropriate variance, and then uses $\theta_0$ to stick with a gradient mapping $\phi(\cdot)$ which will be kept frozen from beginning to end. The algorithm also takes as input the complexity parameter $S = S_{T,n}(h)$ of the underlying function $h$ satisfying (1). We shall later on remove the assumption of the prior knowledge of $S_{T,n}(h)$. In particular, removing the latter, turns out to be quite challenging from a technical standpoint, and gives rise to a complex online model selection algorithms for active learning in non-parametric regimes.

At each round $t$, Algorithm 1 receives an instance $x_t \in \mathcal{X}$, and constructs the two augmented vectors $x_{t,1} = (x_t, 0)$ and $x_{t,-1} = (0, x_t)$ (intuitively corresponding to the two "actions" of a contextual bandit algorithm). The algorithm predicts the label $y_t$ associated with $x_t$ by maximizing over $a \in \mathcal{Y}$ an upper confidence index $U_{t,a}$ stemming from the linear approximation $h(x_{t,a}) \approx \sqrt{m}\langle \phi(x_{t,a}), \theta_{-1} - \theta_0 \rangle$ subject to ellipsoidal constraints $\mathcal{C}_{t-1}$, as in standard contextual bandit algorithms operating with the frozen mapping $\phi(\cdot)$. In addition, in order to decide whether or not to query label $y_t$, the algorithm estimates its own uncertainty by checking to what extent $U_{t,a_t}$ is close to $1/2$. This uncertainty level is ruled by the time-varying threshold $B_t$, which is expected to shrink to $0$ as time progresses. Notice that $B_t$ is a function of $\gamma_{t-1}$, which in turn includes in its definition the complexity parameter $S$. Finally, if $y_t$ is revealed, the algorithm updates its least-squares estimator $\theta_t$ by a rank-one adjustment of matrix $Z_t$ and an additive update to the bias vector $b_t$. No update is taking place if the label is not queried. The following is our initial building block.[2]

**Theorem 1.** *Let Algorithm 1 be run with parameters $\delta$, $S$, $m$, and $n$ on an i.i.d. sample $(x_1, y_1), \ldots, (x_T, y_T) \sim \mathcal{D}$, where the marginal distribution $\mathcal{D}_\mathcal{X}$ fulfills the low-noise condition with exponent $\alpha \geq 0$ w.r.t. a function $h$ that satisfies (1) and such that $\sqrt{2}S_{T,n}(h) \leq S$. If $m = poly(T, n, \lambda_0^{-1}, \log(1/\delta))$, then with probability at least $1 - \delta$ the cumulative regret $R_T$ and*

---

[2]All proofs are in the appendix.

---

**Algorithm 1:** Frozen NTK Selective Sampler.

---

**Input:** Confidence level $\delta$, complexity parameter $S$, network width $m$, and depth $n$ .

**Initialization:**

- Generate each entry of $W_k$ independently from $\mathcal{N}(0, 2/m)$, for $k \in [n-1]$, and each entry of $W_n$ independently from $\mathcal{N}(0, 1/m)$;

- Define $\phi(x) = g(x; \theta_0)/\sqrt{m}$, where $\theta_0 = \langle W_1, \ldots, W_n \rangle \in \mathbb{R}^p$ is the (frozen) weight vector of the neural network so generated;

- Set $Z_0 = I \in \mathbb{R}^{p \times p}$, $b_0 = 0 \in \mathbb{R}^p$ .

**for** $t = 1, 2, \ldots, T$

    Observe instance $x_t \in \mathcal{X}$ and build $x_{t,a} \in \mathcal{X}^2$, for $a \in \mathcal{Y} = \{-1, +1\}$

    Set $\mathcal{C}_{t-1} = \{\theta : \|\theta - \theta_{t-1}\|_{Z_{t-1}} \leq \frac{\gamma_{t-1}}{\sqrt{m}}\}$, with $\gamma_{t-1} = \sqrt{\log \det Z_{t-1} + 2\log(1/\delta)} + S$

    Set

$$U_{t,a} = \sqrt{m} \max_{\theta \in \mathcal{C}_{t-1}} \langle \phi(x_{t,a}), \theta - \theta_0 \rangle = \sqrt{m} \langle \phi(x_{t,a}), \theta_{t-1} - \theta_0 \rangle + \gamma_{t-1} \|\phi(x_{t,a})\|_{Z_{t-1}^{-1}}$$

    Predict $a_t = \arg\max_{a \in \mathcal{Y}} U_{t,a}$

    Set $I_t = \mathbb{1}\{|U_{t,a_t} - 1/2| \leq B_t\} \in \{0, 1\}$     with     $B_t = B_t(S) = 2\gamma_{t-1} \|\phi(x_{t,a_t})\|_{Z_{t-1}^{-1}}$

    **if** $I_t = 1$

        Query $y_t \in \mathcal{Y}$, and set loss $\ell_t = \ell(a_t, y_t)$

        Update

$$Z_t = Z_{t-1} + \phi(x_{t,a_t})\phi(x_{t,a_t})^\top$$
$$b_t = b_{t-1} + (1 - \ell_t)\phi(x_{t,a_t})$$
$$\theta_t = Z_t^{-1} b_t / \sqrt{m} + \theta_0$$

    **else** $Z_t = Z_{t-1}$, $b_t = b_{t-1}$, $\theta_t = \theta_{t-1}$, $\gamma_t = \gamma_{t-1}$, $\mathcal{C}_t = \mathcal{C}_{t-1}$ .

---

*the total number of queries $N_T$ are simultaneously upper bounded as follows:*

$$R_T = O\left( L_H^{\frac{\alpha+1}{\alpha+2}} \left( L_H + \log(1/\delta) + S^2 \right)^{\frac{\alpha+1}{\alpha+2}} T^{\frac{1}{\alpha+2}} + \log(\log T/\delta) \right)$$

$$N_T = O\left( L_H^{\frac{\alpha}{\alpha+2}} \left( L_H + \log(1/\delta) + S^2 \right)^{\frac{\alpha}{\alpha+2}} T^{\frac{2}{\alpha+2}} + \log(\log T/\delta) \right),$$

*where $L_H = \log \det(I + H)$, $H$ being the NTK matrix of depth $n$ over the set of points* $\{x_{t,a}\}_{t=1,\ldots,T,\, a=\pm 1}$.

The above bounds depend, beyond time horizon $T$, on three relevant quantities: the noise level $\alpha$, the complexity parameters $S$ and the log-determinant quantity $L_H$. Notice that, whereas $S$ essentially quantifies the complexity of the function $h$ to be learned, $L_H$ measures instead the complexity of the NTK itself, hence somehow quantifying the complexity of the function space we rely upon in learning $h$. It is indeed instructive to see how the bounds in the above theorem vary as a function of these quantities. First, as expected, when $\alpha = 0$ we recover the usual regret guarantee $R_T = O(\sqrt{T})$, more precisely a bound of the form $R_T = O((L_H + \sqrt{L_H}S)\sqrt{T})$, with the trivial label complexity $N_T = O(T)$. At the other extreme, when $\alpha \to \infty$ we obtain the guarantees $R_T = N_T = O(L_H(L_H + S^2))$. In either case, if $h$ is "too complex" when projected onto the data, that is, if $S_{T,n}^2(h) = \Omega(T)$, then all bounds become vacuous.[3] At the opposite end of the spectrum, if $\{h\}$ is simple, like a class of linear functions with bounded norm in a $d$-dimensional space, and the network depth $n$ is 2 then $S_{T,n}(h) = O(1)$, and $L_H = O(d \log T)$, hence recovering the rates reported in [16] for the linear case. The quantity $L_H$ is tightly related to the decaying rate of the eigenvalues of the NTK matrix $H$, and is poly-logarithmic in $T$ in several important cases [42]. One relevant example is discussed in [43], which relies on the spectral characterization of NTK in [7, 8]: If $n = 2$ and all points $x^{(i)}$ concentrate on a $d_0$-dimensional subspace of the RKHS spanned by the NTK, then $L_H = O(d_0 \log T)$.

---

[3]The same happens, e.g., to the regret bounds in [45].

It is also important to stress that, via a standard online-to-batch conversion, the result in Theorem 1 can be turned to a compelling guarantee in a traditional statistical learning setting, where the goal is to come up at the end of the $T$ rounds with a hypothesis $f$ whose population loss $L(f) = \mathbb{E}_{x \sim D_\mathcal{X}}[L(f \mid x)]$ exceeds the Bayes optimal population loss $\mathbb{E}_{x_t \sim D_\mathcal{X}}[h(x_{t,a_t^*})] = \mathbb{E}_{x_t \sim D_\mathcal{X}}[\max\{h(x_{t,1}), h(x_{t,-1})\}]$ by a vanishing quantity. Following [16], this online-to-batch algorithm will simply run Algorithm 1 by sweeping over the sequence $\{(x_t, y_t)\}_{t=1,\dots,T}$ only once, and pick one function uniformly at random among the sequence of predictors generated by Algorithm 1 during its online functioning, that is, among the sequence $\{U_t(x)\}_{t=1,\dots,T}$, where $U_t(x) = \arg\max_{a \in \mathcal{Y}} \max_{\theta \in \mathcal{C}_{t-1}} \langle \phi(x_{\cdot,a}), \theta - \theta_0 \rangle$, with $x_{\cdot,1} = (x, 0)$ and $x_{\cdot,-1} = (0, x)$. This randomized algorithm enjoys the following high-probability excess risk guarantee:[4]

$$
\mathbb{E}_{t \sim \text{unif}(T)}[L(U_t)] - \mathbb{E}_{x_t \sim D_\mathcal{X}}[h(x_{t,a_t^*})] = O\left( \left( \frac{L_H\left(L_H + \log(1/\delta) + S^2\right)}{T} \right)^{\frac{\alpha+1}{\alpha+2}} + \frac{\log\log(T/\delta)}{T} \right).
$$

Combining with the guarantee on the number of labels $N_T$ from Theorem 1 (and disregarding log factors), this allows us to conclude that the above excess risk can be bounded as a function of $N_T$ as

$$
\left( \frac{L_H(L_H + S^2)}{N_T} \right)^{\frac{\alpha+1}{2}}, \tag{2}
$$

where $L_H(L_H + S^2)$ plays the role of a (compound) complexity term projected onto the data $x_1, \dots, x_T$ at hand. When restricting to VC-classes, the convergence rate $N_T^{-\frac{\alpha+1}{2}}$ is indeed the best rate (*minimax* rate) one can achieve under the Mammen-Tsybakov low-noise condition with exponent $\alpha$ (see, e.g., [10, 20, 27, 16]).

Yet, since we are not restricting to the parametric case, both $L_H$ and, more importantly, $S^2$ can be a function of $T$. In such cases, the generalization bound in (2) can still be expressed as a function of $N_T$ alone, For instance, when $L_H$ is poly-logarithmic in $T$ and $S^2 = O(T^\beta)$, for some $\beta \in [0, 1)$, one can easily verify that (2) takes the form $N_T^{-\frac{(1-\beta)(\alpha+1)}{2+\beta\alpha}}$ (again, up to log factors).

In Section A.3 of the appendix, we extend all our results to the case where the network weights are not frozen, but are updated on the fly according to a gradient descent procedure. In this case, in Algorithm 1 the gradient vector $\phi(x) = g(x; \theta_0)/\sqrt{m}$ will be replaced by $\phi_t(x) = g(x; \theta_{t-1})/\sqrt{m}$, where $\theta_t$ is not the linear-least squares estimator $\theta_t = Z_t^{-1} b_t/\sqrt{m} + \theta_0$, as in Algorithm 1, but the result of the DNN training on the labeled data $\{(x_k, y_k) : k \leq t, I_k = 1\}$ gathered so far.

## 4 Model Selection

Our model selection algorithm is described in Algorithm 2. The algorithm operates on a pool of *base learners* of Frozen NTK selective samplers like those in Algorithm 1, each member in the pool being parametrized by a pair of parameters $(S_i, d_i)$, where $S_i$ plays the role of the (unknown) complexity parameter $S_{T,n}(h)$ (which was replaced by $S$ in Algorithm 1), and $d_i$ plays the role of an (a-priori unknown) upper bound on the relevant quantity $\sum_{t \in T : i_t = i} \frac{1}{2} \wedge I_{t,i} B_{t,i}^2$ that is involved in the analysis (see Lemma 5 and Lemma 7 in Appendix A.1). This quantity will at the end be upper bounded by a term of the form $L_H(L_H + \log(T/\delta) + S_{T,n}^2(h))$ whose components $L_H$ and $S_{T,n}^2(h)$ are initially unknown to the algorithm.

Algorithm 2 maintains over time a set $\mathcal{M}_t$ of active base learners, and a probability distribution $\boldsymbol{p}_t$ over them. This distribution remains constant throughout a sequence of rounds between one change to $\mathcal{M}_t$ and the next. We call such sequence of rounds an *epoch*. Upon observing $x_t$, Algorithm 2 selects which base learner to rely upon in issuing its prediction $a_t$ and querying the label $y_t$, by drawing base learner $i_t \in \mathcal{M}_t$ according to $\boldsymbol{p}_t$.

Then Algorithm 2 undergoes a series of carefully designed elimination tests which are meant to rule out mis-specified base learners, that is, those whose associated parameter $S_i$ is likely to be smaller than $S_{T,n}(h)$, while retaining those such that $S_i \geq S_{T,n}(h)$. These tests will help keep both the regret bound and the label complexity of Algorithm 2 under control. Whenever, at the end of some

---

[4]Observe that this is a *data-dependent* bound, in that the RHS is random variable. This is because both $L_H$ and $S$ may depend on $x_1, \dots, x_T$.

round $t$, any such test triggers, that is, when it happens that $|\mathcal{M}_{t+1}| < |\mathcal{M}_t|$ at the end of the round, a new epoch begins, and the algorithm starts over with a fresh distribution $\boldsymbol{p}_{t+1} \neq \boldsymbol{p}_t$.

The first test ("disagreement test") restricts to all active base learners that would not have requested the label if asked. As our analysis for the base selective sampler (see Lemma 8 in Appendix A.1) shows that a well-specified base learner does not suffer (with high probability) any regret on non-queried rounds, any disagreement among them reveals mis-specification, thus we eliminate in pairwise comparison the base learner that holds the smaller $S_i$ parameter. The second test ("observed regret test") considers the regret behavior of each pair of base learners $i, j \in \mathcal{M}_t$ on the rounds $k \leq t$ on which $i$ was selected ($i_k = i$) and requested the label ($I_{k,i} = 1$), but $j$ would not have requested if asked ($I_{k,j} = 0$), and the predictions of the two happened to disagree on that round ($a_{k,i} \neq a_{k,j}$). The goal here is to eliminate base learners whose cumulative regret is likely to exceed the regret of the smallest well-specified learner, while ensuring (with high probability) that any well-specified base learner $i$ is not removed from the pool. In a similar fashion, the third test ("label complexity test") is aimed at keeping under control the label complexity of the base learners in the active pool $\mathcal{M}_t$. Finally, the last test ("$d_i$ test") simply checks whether or not the candidate value $d_i$ associated with base learner $i$ remains a valid (and tight) upper bound on $L_H(L_H + S_{T,n}^2(h))$.

Notice that the sampling distribution $\boldsymbol{p}_t$ plays base learners with small $d_i$ more often than learners with large $d_i$. Note also that $d_i$ is exactly the (instance-dependent) factor in the cumulative regret and label complexity bounds for base learners that are well-specified. This means that base learners with lower regret are chosen more frequently than base learners that accumulate regret quicker (and similarly for label complexity). In fact, the sampling distribution is chosen so that the total contribution to the cumulative regret of each base learner is roughly equal. As a consequence, the total cumulative regret of Algorithm 2 is at most $M$ (number of base learners) times the regret of each base learner, and the best base learner in particular, which is a key property for achieving the guarantees in Theorem 2 below. Of course, this only works when the base learners are well-specified but the four tests in Algorithm 2 ensure that all other learners are eventually eliminated.

We have the following result, whose proof is contained in Appendix A.2.

**Theorem 2.** *Let Algorithm 2 be run with parameters $\delta$, $\gamma \leq \alpha$ with a pool of base learners $\mathcal{M}_1$ of size $M$ on an i.i.d. sample $(x_1, y_1), \ldots, (x_T, y_T) \sim \mathcal{D}$, where the marginal distribution $\mathcal{D}_\mathcal{X}$ fulfills the low-noise condition with exponent $\alpha \geq 0$ w.r.t. a function $h$ that satisfies (1) and having complexity $S_{T,n}(h)$. Let also $\mathcal{M}_1$ contain at least one base learner $i$ such that $\sqrt{2}S_{T,n}(h) \leq S_i \leq 2\sqrt{2}S_{T,n}(h)$ and $d_i = \Theta(L_H(L_H + \log(M/\delta) + S_{T,n}^2(h)))$, where $L_H = \log \det(I + H)$, being $H$ the NTK matrix of depth $n$ over the set of points $\{x_{t,a}\}_{t=1,\ldots,T,\, a=\pm 1}$. If $m = poly(T, n, \lambda_0^{-1}, \log(1/\delta))$, then with probability at least $1 - \delta$ the cumulative regret $R_T$ and the total number of queries $N_T$ are simultaneously upper bounded as follows:*

$$R_T = O\left( M \left( L_H\big(L_H + \log(M/\delta) + S_{T,n}^2(h)\big) \right)^{\gamma+1} T^{\frac{1}{\gamma+2}} + M\, L(T, \delta) \right)$$

$$N_T = O\left( M \left( L_H\big(L_H + \log(M/\delta) + S_{T,n}^2(h)\big) \right)^{\frac{\gamma}{\gamma+2}} T^{\frac{2}{\gamma+2}} + M\, L(T, \delta) \right),$$

*where $L(T, \delta)$ is the logarithmic term defined at the beginning of Algorithm 2's pseudocode.*

We run Algorithm 2 with the pool $\mathcal{M}_1 = \{(S_{i_1}, d_{i_2})\}$, where $S_{i_1} = 2^{i_1}$, $i_1 = 0, 1, \ldots, O(\log T)$ and $d_{i_2} = 2^{i_2}$, $i_2 = 0, 1, \ldots, O(\log T)$, ensuring[5] the existence of a pair $(i_1, i_2)$ such that

$$\sqrt{2}S_{T,n}(h) \leq S_{i_1} \leq 2\sqrt{2}S_{T,n}(h)$$

and

$$L_H\big(L_H + \log(M/\delta) + S_{T,n}^2(h)\big) \leq d_{i_2} \leq 2L_H\big(L_H + \log(M/\delta) + S_{T,n}^2(h)\big) .$$

Hence the resulting error due to the discretization is just a constant factor, while the resulting number $M$ of base learners is $O(\log^2 T)$.

Theorem 2 allows us to conclude that running Algorithm 2 on the above pool of copies of Algorithm 1 yields guarantees that are similar to those obtained by running a single instance of Algorithm 1 with

---

[5]Notice that the bounds in Theorem 2 become vacuous if either $S_{T,n}(h)$ or $L_H$ are $\Theta(\sqrt{T})$, hence we are only interested in making indices $i_1$ and $i_2$ reach a value which is at most logarithmic in $T$.

**Algorithm 2:** Frozen NTK Selective Sampler with Model Selection.

---

**Input:** Confidence level $\delta$; probability parameter $\gamma \geq 0$; pool of base learners $\mathcal{M}_1$, each identified with a pair $(S_i, d_i)$; number of rounds $T$.

Set $L(t, \delta) = \log \frac{5.2 \log(2t)^{1.4}}{\delta}$

**for** $t = 1, 2, \ldots, T$

    Observe instance $x_t \in \mathcal{X}$ and build $x_{t,a} \in \mathcal{X}^2$, for $a \in \mathcal{Y} = \{-1, +1\}$

    **for** $i \in \mathcal{M}_t$

        Set $I_{t,i} \in \{0, 1\}$ as the indicator of whether base learner $i$ *would* ask for label on $x_t$

        Set $a_{t,i} \in \mathcal{Y}$ as the prediction of base learner $i$ on $x_t$

        Let $B_{t,i} = B_{t,i}(S_i)$ denote the query threshold of base learner $i$ (from Algorithm 1)

    Select base learner $i_t \sim \boldsymbol{p}_t = (p_{t,1}, p_{t,2}, \ldots, p_{t,|\mathcal{M}_t|})$, where

$$
p_{t,i} = \begin{cases} \dfrac{d_i^{-(\gamma+1)}}{\sum_{j \in \mathcal{M}_t} d_j^{-(\gamma+1)}}, & \text{if } i \in \mathcal{M}_t \\ 0, & \text{otherwise} \end{cases}
$$

    Predict $a_t = a_{t,i_t}$

    **if** $I_{t,i_t} = 1$

        Query label $y_t \in \mathcal{Y}$ and send $(x_t, y_t)$ to base learner $i_t$

    $\mathcal{M}_{t+1} = \mathcal{M}_t$

    Set $\mathcal{N}_t = \{i \in \mathcal{M}_t : I_{t,i} = 0\}$          // (1) Disagreement test

    **for** *all pairs of base learners* $i, j \in \mathcal{N}_t$ *that disagree in their prediction* $(a_{t,i} \neq a_{t,j})$

        Eliminate all learners with smaller $S$:    $\mathcal{M}_{t+1} = \{m \in \mathcal{M}_{t+1} : S_m > \min\{S_i, S_j\}\}$

    **for** *all pairs of base learners* $i, j \in \mathcal{M}_t$          // (2) Observed regret test

        Consider rounds where the chosen learner $i$ requested the label but $j$ did not, and $i$ and $j$ disagree in their prediction:

$$
\mathcal{V}_{t,i,j} = \{k \in [t] : i_k = i, I_{k,i} = 1, I_{k,j} = 0, a_{k,i} \neq a_{k,j}\}
$$

        **if** $\displaystyle\sum_{k \in \mathcal{V}_{t,i,j}} \left( \mathbb{1}\{a_{k,i} \neq y_k\} - \mathbb{1}\{a_{k,j} \neq y_k\} \right) > \sum_{k \in \mathcal{V}_{t,i,j}} (1 \wedge B_{k,i}) + 1.45 \sqrt{|\mathcal{V}_{t,i,j}| L(|\mathcal{V}_{t,i,j}|, \delta)}$

            Eliminate base learner $i$:    $\mathcal{M}_{t+1} = \mathcal{M}_{t+1} \setminus \{i\}$

    **for** $i \in \mathcal{M}_t$          // (3) Label complexity test

        Consider rounds where base learner $i$ was played: $\mathcal{T}_{t,i} = \{k \in [t] : i_k = i\}$

        **if**

$$
\sum_{k \in \mathcal{T}_{t,i}} I_{k,i} > \inf_{\epsilon \in (0, 1/2]} \left( 3\epsilon^{\gamma} |\mathcal{T}_{t,i}| + \frac{1}{\epsilon^2} \sum_{k \in \mathcal{T}_{t,i}} I_{k,i} B_{k,i}^2 \wedge \frac{1}{4} \right) + 2L(|\mathcal{T}_{t,i}|, \delta/(M \log_2(12t)))
$$

            Eliminate base learner $i$:    $\mathcal{M}_{t+1} = \mathcal{M}_{t+1} \setminus \{i\}$

    **for** $i \in \mathcal{M}_t$          // (4) $d_i$ test

        **if** $\sum_{k \in \mathcal{T}_{t,i}} \left( \frac{1}{2} \wedge I_{k,i} B_{k,i}^2 \right) > 8d_i$

            Eliminate base learner $i$:    $\mathcal{M}_{t+1} = \mathcal{M}_{t+1} \setminus \{i\}$

---

$S = \sqrt{2} S_{T,n}(h)$, that is, as if the complexity parameter $S_{T,n}(h)$ were known beforehand. Yet, this model selection guarantee comes at a price, since Algorithm 2 needs to receive as input the noise exponent $\alpha$ (through parameter $\gamma \leq \alpha$) in order to correctly shape its label complexity test.

The very same online-to-batch conversion mentioned in Section 3 can be applied to Algorithm 2. Again, combining with the bound on the number of labels and disregarding log factors, this gives us a high probability excess risk bound of the form

$$
\left( \frac{\left[ L_H \left( L_H + S_{T,n}^2(h) \right) \right]^{\frac{3\alpha+2}{\alpha+2}}}{N_T} \right)^{\frac{\alpha+1}{2}}, \tag{3}
$$

provided $\gamma = \alpha$. Following the same example as at the end of Section 3, when $L_H$ is poly-logarithmic in $T$ and $S^2 = O(T^\beta)$, for some $\beta \in [0, 1)$, one can verify that (3) is of the form $N_T^{-\frac{(1-\beta(\alpha+1))(\alpha+1)}{2+\beta\alpha}}$ (up to log factors), which converges for $\beta < 1/(\alpha + 1)$. Hence, compared to (2) we can ensure convergence in a more restricted set of cases.

Section A.3 in the appendix contains the extension of our model selection procedure to the case where the network weights are themselves updated.

## 5    Conclusions and Work in Progress

We have presented a rigorous analysis of selective sampling and active learning in general non-parametric scenarios, where the complexity of the Bayes optimal predictor is evaluated on the data at hand as a fitting measure with respect to the NTK matrix of a given depth associated with the same data. This complexity measure plays a central role in the level of uncertainty the algorithm assigns to labels (the higher the complexity the higher the uncertainty, hence the more labels are queried). Yet, since this is typically an unknown parameter of the problem, special attention is devoted to designing and analyzing a model selection technique that adapts to this unknown parameter.

In doing so, we borrowed tools and techniques from Neural Bandits [45, 43], selective sampling (e.g., [16]), and online model selection in contextual bandits [36, 35], and combined them together in an original and non-trivial manner.

We proved regret and label complexity bounds that recover known minimax rates in the parametric case, and extended such results well beyond the parametric setting achieving favorable guarantees that cannot easily be compared to available results in the literature of active learning in non-parametric settings. One distinctive feature of our proposed technique is that it gives rise to efficient and manageable algorithms for modular DNN architecture design and deployment.

We conclude by mentioning a few directions we are currently exploring:

1. We are trying to get rid of the prior knowledge of $\alpha$ in the model selection Algorithm 2. This may call for a slightly more refined balancing technique that jointly involves $S_{T,n}(h)$ and $\alpha$ itself.

2. Regardless of whether $\alpha$ is available, it would be nice to improve the dependence on $\gamma = \alpha$ in the regret bound of Theorem 2. This would ensure convergence of the generalization bound as $N_T \to \infty$ when $S_{T,n}(h)^2 = T^\beta$, for all $\beta \in [0, 1)$. We conjecture that this is due to a suboptimal design of our balancing mechanism for model selection in Algorithm 2.

3. We are investigating links between the complexity measure $S_{T,n}(h)$ and the smoothness properties of the (Bayes) regression function $h$ with respect to the NTK kernel (of a given depth $n$).

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
