# A   Appendix

This appendix contains, beyond the proof of all results contained in the main body (Section A.1 and Section A.2), the extension of our model selection results to the non-frozen NTK case (Section A.3). Section A.4 contains ancillary technical lemmas used throughout the proofs.

## A.1   Proofs for Section 3

We first recall the following representation theorem (which is Lemma 5.1 in [45]). We give a proof sketch for completeness.

**Lemma 1.** *There exists a positive constant $C$ such that for any $\delta \in (0,1)$, if*

$$m \geq CT^4 n^6 \log(2Tn/\delta)/\lambda_0^4$$

*then with probability at least $1 - \delta$ over the random initialization $\theta_0$, there exists $\theta^* \in \mathbb{R}^p$ for which*

$$h(x_{t,a}) = \langle g(x_{t,a}; \theta_0), \theta^* - \theta_0 \rangle \qquad \text{and} \qquad \sqrt{m}\,\|\theta^* - \theta_0\|_2 \leq \sqrt{2} S_{T,n}(h) \qquad (4)$$

*for all $t \in [T]$, $a \in \mathcal{Y}$, and $h$.*

*Proof.* Recall the rearrangement of $\{x_{t,a}\}_{t=1,\dots,T,\,a=\pm 1}$ into $\{x^{(i)}\}_{i=1,\dots,2T}$. We define the $p \times 2T$ matrix $G = \left[ \phi(x^{(1)}), \dots, \phi(x^{(2T)}) \right]$. For $m = \Omega(T^4 n^6 \log(2Tn/\delta)/\lambda_0^4)$, we have $\|G^\top G - H\|_F \leq \lambda_0/2$ with probability at least $1 - \delta$ over the random initialization over $\theta_0$, which is based on a union bound over Theorem 3.1 in [2]. Since $H$ on $\{x^{(i)}\}_{i=1,\dots,2T}$ is positive definite with smallest eigenvalue $\lambda_0$, $G^\top G$ is also positive definite. Let the singular value decomposition of $G$ be $G = PAQ^\top$, $P \in \mathbb{R}^{p \times 2T}$, $A \in \mathbb{R}^{2T \times 2T}$, $Q \in \mathbb{R}^{2T \times 2T}$, then $A$ is also positive definite. We define

$$\theta^* = \theta_0 + PA^{-1}Q^\top \mathbf{h}/\sqrt{m}\,.$$

It is easy to see that $\theta^*$ satisfies (4), hence concluding the proof. $\qquad \square$

Next we present a lemma relating the matrix $Z_T$ with NTK matrix $H$.

**Lemma 2.** *There exists a positive constant $C$ such that for any $\delta \in (0,1)$, if*

$$m \geq CT^6 n^6 \log(Tn/\delta)$$

*then with probability at least $1 - \delta$ over the random initialization $\theta_0$ we have*

$$\log \det Z_T \leq \log \det(I + H) + 1\,. \qquad (5)$$

*Proof.* The proof is an adaptation of the proof of Lemma 5.4 in [45]. Let $G = (\phi(x^{(1)}), \dots, \phi(x^{(2T)})) \in \mathbb{R}^{p \times 2T}$. We can write

$$
\begin{aligned}
\log \det Z_T &= \log \det \left( I + \sum_{t=1}^{T} I_t \phi(x_{t,a_t}) \phi(x_{t,a_t})^\top \right) \\
&\leq \log \det \left( I + \sum_{i=1}^{2T} \phi(x^{(i)}) \phi(x^{(i)})^\top \right) \\
&= \log \det \left( I + GG^\top \right) \\
&= \log \det \left( I + G^\top G \right) \\
&= \log \det \left( I + H + (G^\top G - H) \right) \\
&\leq \log \det \left( I + H \right) + \langle (I + H)^{-1}, (G^\top G - H) \rangle_F \\
&\leq \log \det \left( I + H \right) + \|(I + H)^{-1}\|_F \|G^\top G - H\|_F \\
&\leq \log \det \left( I + H \right) + \sqrt{2T}\,\|G^\top G - H\|_F \\
&\leq \log \det (I + H) + 1\,.
\end{aligned}
$$

In the above, the first inequality is obvious, the second inequality uses the fact that $\log \det(\cdot)$ is a concave function, the third one used Cauchy-Schwartz inequality, the fourth one comes from $\|(I + H)^{-1}\|_F \leq \|I\|_F = \sqrt{2T}$, and the last inequality uses Lemma B.1 in [45] along with our choice of $m$. $\qquad \square$

The proofs of both Lemma 1 and Lemma 2 rely on controlling the size of $\|G^\top G - H\|_F$, which is small with high probability when $m$ is large enough. Therefore, given

$$m \geq CT^4 \log(2Tn/\delta)n^6 \left(T^2 \vee 1/\lambda_0^4\right) ,$$

we have

$$\mathcal{E}_0 = \{\exists \theta^* \in \mathbb{R}^p \ : \ (4) \text{ and } (5) \text{ hold}\} , \tag{6}$$

holds with probability at least $1 - \delta$ over random initialization of $\theta_0$.

To take into account the random noise from the sequence of labels, we also define

$$\mathcal{E} = \{\exists \theta^* \in \mathbb{R}^p \ : \ \mathcal{E}_0 \text{ holds and } \theta^* \in \mathcal{C}_t \ \forall t > 0\} . \tag{7}$$

In order to make sense of the querying threshold $B_t$ in Algorithm 1, we derive an upper and a lower bound for $U_{t,a} - h(x_{t,a})$ under $\mathcal{E}$.

As for the lower bound, simply notice that, by definition ,

$$U_{t,a} = \max_{\theta \in \mathcal{C}_{t-1}} \langle g(x_{t,a}; \theta_0), \theta - \theta_0 \rangle \geq \langle g(x_{t,a}; \theta_0), \theta^* - \theta_0 \rangle = h(x_{t,a}) . \tag{8}$$

To derive an upper bound, we can write

$$\begin{aligned}
U_{t,a} - h(x_{t,a}) &= \max_{\theta \in \mathcal{C}_{t-1}} \langle g(x_{t,a}; \theta_0), \theta - \theta_0 \rangle - \langle g(x_{t,a}; \theta_0), \theta^* - \theta_0 \rangle \\
&= \max_{\theta \in \mathcal{C}_{t-1}} \langle g(x_{t,a}; \theta_0), \theta - \theta_{t-1} \rangle - \langle g(x_{t,a}; \theta_0), \theta^* - \theta_{t-1} \rangle \\
&\leq \max_{\theta \in \mathcal{C}_{t-1}} \|g(x_{t,a}; \theta_0)\|_{Z_{t-1}^{-1}} \left( \|\theta - \theta_{t-1}\|_{Z_{t-1}} + \|\theta^* - \theta_{t-1}\|_{Z_{t-1}} \right) \\
&\leq 2\gamma_{t-1} \|\phi(x_{t,a})\|_{Z_{t-1}^{-1}} ,
\end{aligned} \tag{9}$$

where in the last inequality we used the definition of $\mathcal{C}_{t-1}$ and the assumption that $\theta^* \in \mathcal{C}_{t-1}$. A proof of this assumption is contained in the below lemma, which follows from standard arguments.

**Lemma 3.** *Let the input parameter $S$ in Algorithm 1 be such that $\sqrt{2}S_{T,n}(h) \leq S$, then under event $\mathcal{E}_0$ for any $\delta > 0$, with probability at least $1 - \delta$ over the random noises we have*

$$\|\theta^* - \theta_t\|_{Z_t} \leq \gamma_t/\sqrt{m}$$

*for all $t \geq 0$ simultaneously, i.e., $\theta^* \in \mathcal{C}_t$ with high probability simultaneously for all $t \geq 0$.*

*Proof.* We essentially follow the proof of Theorem 2 in [1] (see also the proof of Lemma 5.2 in [45]).

We have $\ell_t = 1 - h(x_{t,a_t}) - \xi_t$, where $\xi_t = 1 - \ell_t - h(x_{t,a_t})$ is a sub-Gaussian random variable. Hence, setting $\boldsymbol{\xi}_t = (I_1 \xi_1, ..., I_t \xi_t)^\top$, $X_t = (I_1 \phi(x_{1,a_1}), ..., I_t \phi(x_{t,a_t}))^\top$, and $Y_t = (I_1(1 - \ell_1), ..., I_t(1 - \ell_t))^\top$, we can write

$$Z_t = X_t^\top X_t + I, \qquad b_t = X_t^\top Y_t$$

Plug them into the definition of $\theta_t$ gives

$$\begin{aligned}
\theta_t - \theta_0 &= Z_t^{-1} b_t/\sqrt{m} \\
&= (X_t^\top X_t + I)^{-1} X_t^\top (\sqrt{m} X_t(\theta^* - \theta_0) + \boldsymbol{\xi}_t)/\sqrt{m} \\
&= (X_t^\top X_t + I)^{-1} X_t^\top \boldsymbol{\xi}_t/\sqrt{m} + \theta^* - \theta_0 - (X_t^\top X_t + I)^{-1}(\theta^* - \theta_0) ,
\end{aligned}$$

where in the first equality we used definition of $\xi_t$ and Lemma 1. Now, for any $x \in \mathbb{R}^p$, we get

$$x^\top(\theta_t - \theta^*) = \langle x, X_t^\top \boldsymbol{\xi}_t \rangle_{Z_t^{-1}}/\sqrt{m} - \langle x, \theta^* - \theta_0 \rangle_{Z_t^{-1}} ,$$

hence

$$\begin{aligned}
|x^\top(\theta_t - \theta^*)| &\leq \|x\|_{Z_t^{-1}} \left( \|X_t^\top \boldsymbol{\xi}_t\|_{Z_t^{-1}}/\sqrt{m} + \|\theta^* - \theta_0\|_{Z_t^{-1}} \right) \\
&\leq \|x\|_{Z_t^{-1}} \left( \|X_t^\top \boldsymbol{\xi}_t\|_{Z_t^{-1}}/\sqrt{m} + \|\theta^* - \theta_0\|_2 \right) ,
\end{aligned}$$

where the first inequality derives from the Cauchy-Schwartz inequality and the second from the fact that the smallest eigenvalue of $Z_t$ is at least 1. Then, by Theorem 1 in [1], for any $\delta$ with probability at least $1 - \delta$ over the random noises

$$\|X_t^\top \boldsymbol{\xi}_t\|_{Z_t^{-1}} \leq \sqrt{\log\left(\frac{\det(Z_t)}{\delta^2}\right)} \ .$$

Therefore, when $\mathcal{E}_0$ holds, we have for all $t > 0$, with probability at least $1 - \delta$,

$$|x^\top (\theta_t - \theta^*)| \leq \|x\|_{Z_t^{-1}} \left( \sqrt{\log\left(\frac{\det(Z_t)}{\delta^2}\right)}/m + \sqrt{2} S_{T,n}(h)/\sqrt{m} \right) \ .$$

Plugging in $x = Z_t(\theta_t - \theta^*)$ and using $\sqrt{2} S_{T,n}(h) \leq S$, we obtain

$$\|\theta^* - \theta_t\|_{Z_t} \leq \sqrt{\log\left(\frac{\det(Z_t)}{\delta^2}\right)/m} + S/\sqrt{m} = \gamma_t/\sqrt{m} \ ,$$

as claimed. □

Combining Lemma 1, 2 and 3 we confirm that $\mathcal{E}$ is a high probability event.

**Lemma 4.** *There exists a constant $C$ such that if $m \geq CT^4 \log(2Tn/\delta)n^6 \left(T^2 \vee 1/\lambda_0^4\right)$ and $\sqrt{2} S_{T,n}(h) \leq S$, then*

$$\mathbb{P}(\mathcal{E}) \geq 1 - 2\delta \ . \tag{10}$$

*Proof.* Lemma 1 and 2 imply that $\mathbb{P}(\mathcal{E}_0) \geq 1 - \delta$ when $m \geq CT^4 \log(2Tn/\delta)n^6 \left(T^2 \vee 1/\lambda_0^4\right)$. Lemma 3 implies that when $\sqrt{2} S_{T,n}(h) \leq S$, $\mathbb{P}(\theta^* \in \mathcal{C}_t \ \forall t > 0 \mid \mathcal{E}_0) \geq 1 - \delta$. Therefore,

$$\mathbb{P}(\mathcal{E}) = \mathbb{P}(\theta^* \in \mathcal{C}_t \ \forall t > 0 \mid \mathcal{E}_0)\mathbb{P}(\mathcal{E}_0) \geq (1 - \delta)^2 \geq 1 - 2\delta \ .$$

□

**Lemma 5.** *For any $b > 0$ we have*

$$\sum_{t=1}^T b \wedge I_t B_t^2 \leq 8 \left( \log \det Z_T + 2\log(1/\delta) + S^2 + \frac{b}{8} \right) \log \det Z_T \ . \tag{11}$$

*Proof.* By definition of $B_t$ and the fact that $\gamma_t$ is increasing, we have

$$\sum_{t=1}^T b \wedge I_t B_t^2 \leq 4\gamma_T^2 \sum_{t=1}^T \frac{b}{4\gamma_T^2} \wedge I_t \|\phi(x_{t,a_t})\|_{Z_{t-1}^{-1}}^2 \leq (b + 4\gamma_T^2) \log \det Z_T \ ,$$

where the second inequality is from Lemma 24. Using the definition of $\gamma_T$ and the inequality $(a + b)^2 \leq 2a^2 + 2b^2$ we obtain

$$\gamma_T^2 \leq 2\log \det Z_T + 4\log(1/\delta) + 2S^2 \ .$$

Plugging this in we get (11). □

Let us now introduce the short-hand notation

$$\widehat{\Delta}_t = U_{t,a_t} - 1/2 \ , \qquad \Delta_t = h(x_{t,a_t}) - 1/2 \ , \qquad T_\epsilon = \sum_{t=1}^T \mathbb{1}\{\Delta_t^2 \leq \epsilon^2\} \ ,$$

for some $\epsilon \in (0, \frac{1}{2})$. Combined with (8) and (9), we have the following statement about $\widehat{\Delta}_t$ and $\Delta_t$.

**Lemma 6.** *Under event $\mathcal{E}$, $0 \leq \widehat{\Delta}_t - \Delta_t \leq B_t$ and $0 \leq \widehat{\Delta}_t$ hold for all $t$, where $B_t$ is the querying threshold in Algorithm 1, i.e.,*

$$B_t = 2\gamma_{t-1}\|\phi(x_{t,a_t})\|_{Z_{t-1}^{-1}} \ .$$

*Proof.* Recalling that (8) and (9) implies that for $a \in \mathcal{Y}$

$$0 \le U_{t,a} - h(x_{t,a}) \le B_t .$$

Specifically when $a = a_t$,

$$0 \le \widehat{\Delta}_t - \Delta_t \le B_t .$$

Also using (8) we have $U_{t,1} + U_{t,-1} \ge h(x_{t,1}) + h(x_{t,-1}) = 1$. Hence, by definition of $a_t$, $U_{t,a_t} \ge 1/2$, i.e., $\widehat{\Delta}_t \ge 0$. $\qquad \square$

The following lemma bounds the label complexity $N_T$ of Algorithm 1 under event $\mathcal{E}$. Notice that, as stated, the bound does not depend on any specific properties of the marginal distribution $\mathcal{D}_{\mathcal{X}}$.

**Lemma 7.** *Under event $\mathcal{E}$, for any $\epsilon \in (0, 1/2)$ we have*

$$N_T \le T_\epsilon + \frac{8}{\epsilon^2}(\log \det Z_T + 2\log(1/\delta) + S^2 + \frac{1}{32}) \log \det Z_T$$

$$= O\left( T_\epsilon + \frac{1}{\epsilon^2} \left(\log \det(I + H) + \log(1/\delta) + S^2\right) \log \det(I + H) \right) .$$

*Proof.* We adapt the proof of Lemma 6 in [16]. Assume $\mathcal{E}$ holds. Since $0 \le \widehat{\Delta}_t - \Delta_t \le B_t$ and $\widehat{\Delta}_t \ge 0$ by Lemma 6, $\widehat{\Delta}_t \le B_t$ implies $|\Delta_t| \le B_t$. We can write

$$I_t = I_t \mathbb{1}\left\{ \widehat{\Delta}_t \le B_t \right\}$$

$$\le I_t \mathbb{1}\left\{ \widehat{\Delta}_t \le B_t, B_t \ge \epsilon \right\} + I_t \mathbb{1}\left\{ \widehat{\Delta}_t \le B_t, B_t < \epsilon \right\}$$

$$\le \frac{I_t B_t^2}{\epsilon^2} \wedge 1 + \mathbb{1}\{\Delta_t^2 \le \epsilon^2\} .$$

For the first term, summing over $t$ yields

$$\frac{1}{\epsilon^2} \sum_{t=1}^{T} I_t B_t^2 \wedge \epsilon^2 \le \frac{1}{\epsilon^2} \sum_{t=1}^{T} I_t B_t^2 \wedge \frac{1}{4}$$

$$\le \frac{8}{\epsilon^2} \left( \log \det Z_T + 2\log(1/\delta) + S^2 + \frac{1}{32} \right) \log \det Z_T$$

$$= O\left( \frac{1}{\epsilon^2} \left(\log \det(I + H) + \log(1/\delta) + S^2\right) \log \det(I + H) \right) ,$$

where the second bound follows from Lemma 5, and the last bound holds under event $\mathcal{E}$. $\qquad \square$

The next lemma shows that on rounds where Algorithm 1 does not issue a query, we are confident that our prediction $a_t$ suffers no regret.

**Lemma 8.** *Under event $\mathcal{E}$, for the rounds $t$ such that $I_t = 0$, we have $a_t = a_t^*$, that is, Algorithm 1 suffers no regret.*

*Proof.* We apply Lemma 6, when $I_t = 0$ this yields $\widehat{\Delta}_t > B_t$. As a consequence of the condition $\widehat{\Delta}_t - \Delta_t \le B_t$, we get $\Delta_t > 0$, which in turn entails $a_t = a_t^*$. $\qquad \square$

The next lemma establishes an upper bound on the cumulative regret $R_T$ in the same style as in Lemma 7.

**Lemma 9.** *Under event $\mathcal{E}$, for any $\epsilon \in (0, 1/2)$ we have*

$$R_T \le 2\epsilon T_\epsilon + \frac{16}{\epsilon}\left( \log \det Z_T + 2\log(1/\delta) + S^2 + \frac{1}{16} \right) \log \det Z_T$$

$$= O\left( \epsilon T_\epsilon + \frac{1}{\epsilon} \left(\log \det(I + H) + \log(1/\delta) + S^2\right) \log \det(I + H) \right) .$$

*Proof.* By virtue of Lemma 8, we can restrict with high probability to the rounds $t$ on which $I_t = 1$. We have

$$
\begin{aligned}
R_T &= \sum_{t=1}^{T} I_t \big( h(x_{t,a_t^*}) - h(x_{t,a_t}) \big) \\
&= \sum_{t=1}^{T} I_t \big( h(x_{t,a_t^*}) - h(x_{t,a_t}) \big) \mathbb{1}\{a_t \neq a_t^*\} \\
&\leq \sum_{t=1}^{T} I_t \big| h(x_{t,1}) - h(x_{t,-1}) \big| \mathbb{1}\{a_t \neq a_t^*\} \\
&= 2 \sum_{t=1}^{T} I_t |\Delta_t| \\
&= 2 \sum_{t=1}^{T} I_t |\Delta_t| \mathbb{1}\{|\Delta_t| > \epsilon\} + 2 \sum_{t=1}^{T} I_t |\Delta_t| \mathbb{1}\{|\Delta_t| \leq \epsilon\} .
\end{aligned}
$$

The second sum is clearly upper bounded by $2\epsilon T_\epsilon$. As for the first sum, notice that Lemma 6 along with $I_t = 1$ implies $|\Delta_t| \leq B_t$ under event $\mathcal{E}$. Therefore

$$
\begin{aligned}
2 \sum_{t=1}^{T} I_t |\Delta_t| \mathbb{1}\{|\Delta_t| > \epsilon\} &\leq \frac{2}{\epsilon} \sum_{t=1}^{T} I_t \Delta_t^2 \wedge \epsilon \\
&\leq \frac{2}{\epsilon} \sum_{t=1}^{T} I_t B_t^2 \wedge \frac{1}{2} \\
&\leq \frac{16}{\epsilon} \left( \log \det Z_T + 2 \log(1/\delta) + S^2 + \frac{1}{16} \right) \log \det Z_T \\
&= O\left( \frac{1}{\epsilon} \big( \log \det(I + H) + \log(1/\delta) + S^2 \big) \log \det(I + H) \right) .
\end{aligned}
$$

The third bound follows from Lemma 5, while the last bound holds under event $\mathcal{E}$. $\qquad\square$

At this point, we leverage the fact that $x_1, ..., x_T$ are generated in an i.i.d. fashion according to a marginal distribution $\mathcal{D}_\mathcal{X}$ satisfying the low-noise assumption with exponent $\alpha$ recalled in Section 3. A direct application of Lemma 23 (Appendix A.4) gives, with probability at least $1 - \delta$,

$$
T_\epsilon \leq 3T\epsilon^\alpha + O\left( \log \frac{\log T}{\delta} \right) ,
$$

simultaneously over $\epsilon$. Using the above bound on $T_\epsilon$ back into both Lemma 7 and Lemma 9 and optimizing over $\epsilon$ in the two bounds separately yields the following result, which is presented in the main body as Theorem 1.

**Theorem 3.** *Let Algorithm 1 be run with parameters $\delta$, $S$, $m$, and $n$ on an i.i.d. sample $(x_1, y_1), \ldots, (x_T, y_T) \sim \mathcal{D}$, where the marginal distribution $\mathcal{D}_\mathcal{X}$ fulfills the low-noise condition with exponent $\alpha \geq 0$ w.r.t. a function $h$ that satisfies (1) and such that $\sqrt{2} S_{T,n}(h) \leq S$ for all $\{x_i\}_{i=1}^{T}$. Also assume $m \geq CT^4 \log(2Tn/\delta)n^6 \left( T^2 \vee 1/\lambda_0^4 \right)$ where $C$ is the constant in Lemma 1 and Lemma 2. Then with probability at least $1 - \delta$ the cumulative regret $R_T$ and the total number of queries $N_T$ are simultaneously upper bounded as follows:*

$$
R_T = O\left( L_H^{\frac{\alpha+1}{\alpha+2}} \left( L_H + \log(1/\delta) + S^2 \right)^{\frac{\alpha+1}{\alpha+2}} T^{\frac{1}{\alpha+2}} + \log(\log T/\delta) \right)
$$

$$
N_T = O\left( L_H^{\frac{\alpha}{\alpha+2}} \left( L_H + \log(1/\delta) + S^2 \right)^{\frac{\alpha}{\alpha+2}} T^{\frac{2}{\alpha+2}} + \log(\log T/\delta) \right) ,
$$

*where $L_H = \log \det(I + H)$, and $H$ is the NTK matrix of depth $n$ over the set of points $\{x_{t,a}\}_{t=1,...,T, a=\pm 1}$.*

## A.2 Proofs for Section 4

**Additional notation.** In this section, we add subscript "$i$" to the relevant quantities occurring in the proof when these quantities refer to the $i$-th base learner. For instance, we write $Z_{t,i}$ to denote the covariance matrix updated within the $i$-th base learner, $B_{t,i} = B_{t,i}(S_i) = 2\gamma_{t-1,i}\|\phi(x_{t,a_t})\|_{Z_{t-1,i}^{-1}}$, with $\gamma_{t-1,i} = \sqrt{\log \det Z_{t-1,i} + 2\log(1/\delta)} + S_i$, and $\mathcal{C}_{t,i}$ to denote the confidence ellipsoid maintained by the $i$-th base learner.

For convenience, we also introduce the function

$$d(S,\delta) = (\log \det(I + H) + 1)(\log \det(I + H) + \frac{17}{16} + 2\log(M/\delta) + S^2). \tag{12}$$

The above is a high probability upper bound on $(\frac{1}{16} + \frac{1}{2}\gamma_{T,i}^2) \log \det Z_{T,i}$ (holding for all $i$), which in turn upper bounds $\frac{1}{8}\sum_{t=1}^{T} I_{t,i}B_{t,i}^2 \wedge \frac{1}{2}$.

By the assumption in Theorem 2, we know that there is a learner $i^\star = \langle i_1^\star, i_2^\star \rangle \in \mathcal{M}_1$ such that its parameters $S_{i_1^\star}$ and $d_{i_2^\star}$ satisfy

$$\sqrt{2}S_{T,n}(h) \leq S_{i_1^\star} \leq 2\sqrt{2}S_{T,n}(h) \tag{13}$$

$$d(S_{T,n}(h),\delta) \leq d(S_{i_1^\star},\delta) \leq d_{i_2^\star} \leq 2d(S_{i_1^\star},\delta) \leq 8d(S_{T,n}(h),\delta). \tag{14}$$

Throughout the proof we will refer to a specific learner that satisfies these conditions by $i^\star$. Moreover, we denote by $\mathcal{E}_i$ the event where the conditions of the event in Eq. (7) and the event in Lemma 2 hold for base learner $i$. In $\mathcal{E}_i$, we call $i$ well-specified.

Let $R(\mathcal{T})$ and $N(\mathcal{T})$ denote cumulative regret $R$ and number of requested labels $N$ when restricted to subset $\mathcal{T} \subseteq [T]$. Then the regret and label complexity analyses of Algorithm 1 in Section A.1 directly imply the following regret and label complexity bounds of a well-specified base learner $i$ during the execution of Algorithm 2.

**Lemma 10** (Regret and label complexity of a well-specified base learner). *Let $i \in \mathcal{M}_1$ be any base learner. In event $\mathcal{E}_i$ (when $i$ is well-specified), the following regret and label complexity bound holds for any $0 < \epsilon < \frac{1}{2}$ and $t \in [T]$:*

$$R(\mathcal{T}_{t,i}) \leq 2\sum_{k \in \mathcal{T}_{t,i}} I_{k,i}B_{k,i} \wedge \frac{1}{2} \leq \frac{16}{\epsilon}d(S_{i_1},\delta) + 2\epsilon|\mathcal{T}_{t,i}^\epsilon|$$

$$N(\mathcal{T}_{t,i}) \leq |\mathcal{T}_{t,i}^\epsilon| + \frac{1}{\epsilon^2}\sum_{k \in \mathcal{T}_{t,i}} I_{k,i}B_{k,i}^2 \wedge \frac{1}{4} \leq \frac{8}{\epsilon^2}d(S_{i_1},\delta) + |\mathcal{T}_{t,i}^\epsilon|,$$

*where $\mathcal{T}_{t,i}^\epsilon = \{k \in [t]: i_k = i, |\Delta_k| \leq \epsilon\}$. Furthermore, in rounds $t \in \mathcal{T}_{t,i}$ where the label is not queried ($I_{t,i} = 0$), the regret is 0.*

*Proof.* This follows directly from the analysis of Algorithm 1 in the previous section. $\square$

Equipped with these two properties of well-specified base learners, we can first show that with high probability, Algorithm 2 will never eliminate a well-specified learner, and subsequently analyze the label complexity and cumulative regret of Algorithm 2.

**Lemma 11.** *Let $i = \langle i_1, i_2 \rangle \in \mathcal{M}_1$ be a base learner with $d_{i_2} \geq d(S_{i_1},\delta)$. Assume $\gamma \leq \alpha$ and consider event $\bigcap_{j: j \geq i_1} \mathcal{E}_j$. Then, under that event, with probability at least $1 - M\delta$ Algorithm 2 never eliminates base learner $i$.*

*Proof.* We show the statement for each of the four mis-specification tests in turn:

- **Disagreement test:** Consider a round $t$ and any learner $j = \langle j_1, j_2 \rangle$ with $S_{j_1} \geq S_{i_1}$ and $I_{t,i} = I_{t,j} = 0$. By assumption, $\mathcal{E}_i \cap \mathcal{E}_j$ holds. Since $i$ did not ask for the label, this implies that $|\Delta_t| > 0$ (since in rounds with no margin $|\Delta_t| = 0$, a learner always asks for the label). Further, by Lemma 10, the prediction of $i$ and $j$ has no regret in round $t$. Thus, $i$ and $j$ need to make the same prediction and the test does not trigger.

- **Observed regret test:** Consider a round $t$ and any $j \in \mathcal{M}_t$. Then, by virtue of Lemma 21 (Appendix A.4), the left-hand side of the observed regret test for pair $(i, j)$ is upper-bounded with probability at east $1 - \delta$ as

$$\sum_{k \in \mathcal{V}_{t,i,j}} (\mathbb{1}\{a_{k,i} \neq y_k\} - \mathbb{1}\{a_{k,j} \neq y_k\})$$

$$\leq \sum_{k \in \mathcal{V}_{t,i,j}} (h(x_{k,a_{k,j}}) - h(x_{k,a_{k,i}})) + 0.72\sqrt{|\mathcal{V}_{t,i,j}| L(|\mathcal{V}_{t,i,j}|, \delta)}$$

$$\leq \sum_{k \in \mathcal{V}_{t,i,j}} (h(x_{k,a_k^\star}) - h(x_{k,a_{k,i}})) + 0.72\sqrt{|\mathcal{V}_{t,i,j}| L(|\mathcal{V}_{t,i,j}|, \delta)}$$

$$= R(\mathcal{V}_{t,i,j}) + 0.72\sqrt{|\mathcal{V}_{t,i,j}| L(|\mathcal{V}_{t,i,j}|, \delta)} \,,$$

where the second inequality follows from the definition of the best prediction $a_k^*$ for round $k$. Finally, in event $\mathcal{E}_i$ the regret of $i$ in rounds $\mathcal{V}_{t,i,j}$ is bounded by Lemma 10 as

$$R(\mathcal{V}_{t,i,j}) \leq \sum_{k \in \mathcal{V}_{t,i,j}} 1 \wedge B_{k,i} \,.$$

Therefore, this test does not trigger for pair $(i, j)$ in round $t$. By a union bound, this happens with probability at least $1 - M\delta$.

- **Label complexity test:** By Lemma 10, the number of labels requested by $i$ up to round $t$ is at most

$$\sum_{k \in \mathcal{T}_{t,i}} I_{k,i} \leq \inf_{\epsilon \in (0,1/2]} |\mathcal{T}_{t,i}^\epsilon| + \frac{1}{\epsilon^2} \sum_{k \in \mathcal{T}_{t,i}} I_{k,i} B_{k,i}^2 \wedge \frac{1}{4} \,.$$

We now use Lemma 23 (Appendix A.4) to upper-bound $|\mathcal{T}_{t,i}^\epsilon|$ simultaneously for all $\epsilon$ as

$$|\mathcal{T}_{t,i}^\epsilon| \leq 3\epsilon^\gamma |\mathcal{T}_{t,i}| + 2L(|\mathcal{T}_{t,i}|, \delta/\log_2(12t)) \,.$$

By plugging this expression into the previous bound (and taking a union bound over $i$) we show that the label complexity test is not triggered.

- $d_i$ **test:** Using the assumption that $\mathcal{E}_i$ holds and Lemma 5, we can bound the left-hand side of the test as

$$\sum_{k \in \mathcal{T}_{t,i}} (\frac{1}{2} \wedge I_{k,i} B_{k,i}^2) \leq 8(\log \det Z_{t,i} + 2\log(1/\delta) + S_{i_1}^2 + 1/16) \log \det Z_{t,i}$$

$$\leq 8(\log \det(H + I) + 2\log(1/\delta) + S_{i_1}^2 + 17/16)(\log \det(H + I) + 1)$$
$$= 8d(S_{i_1}, \delta)$$

and by the assumption that $d_{i_2} \geq d(S_{i_1}, \delta)$, learner $i$ is not be eliminated by this test.

This concludes the proof. $\qquad \square$

### A.2.1    Label Complexity Analysis

**Lemma 12** (Label complexity of Algorithm 2). *In event* $\bigcap_{i = \langle i_1, i_2 \rangle \in \mathcal{M}_1 \,:\, i_1 \geq i_1^\star} \mathcal{E}_i$*, Algorithm 2 queries with probability at least* $1 - M\delta$

$$N(T) = O\left( \sum_{i = \langle i_1, i_2 \rangle \in \mathcal{M}_1} \left( \frac{d_{i_2}}{\epsilon^2} + \epsilon^\gamma T \left( 1 \wedge \frac{d(S_{T,n}(h), \delta)}{d_{i_2}} \right)^{\gamma+1} \right) + ML(T, \delta/\log T) \right)$$

*labels.*

*Proof.* We can decompose the total number of label requests as

$$N(T) = \sum_{t=1}^{T} I_{t,i_t} = \sum_{i=1}^{M} \sum_{t \in \mathcal{T}_{T,i}} I_{t,i} = \sum_{i \in \mathcal{M}_1} N(\mathcal{T}_{T,i}) \, .$$

Since each learner $i$ satisfied the label complexity test except possibly for the round where it was eliminated, we have

$$N(\mathcal{T}_{T,i}) = O\left( \inf_{\epsilon \in (0,1/2)} \left( \epsilon^{\gamma} |\mathcal{T}_{T,i}| + \frac{1}{\epsilon^2} \sum_{k \in \mathcal{T}_{t,i}} I_{k,i} B_{k,i}^2 \wedge \frac{1}{4} \right) + L(|\mathcal{T}_{T,i}|, \delta/\log t) \right)$$

$$= O\left( \inf_{\epsilon \in (0,1/2)} \left( \epsilon^{\gamma} \sum_{k \in [T]} p_{k,i} + \frac{1}{\epsilon^2} \sum_{k \in \mathcal{T}_{t,i}} I_{k,i} B_{k,i}^2 \wedge \frac{1}{4} \right) + L(T, \delta/\log T) \right)$$

$$= O\left( \inf_{\epsilon \in (0,1/2)} \left( \epsilon^{\gamma} \sum_{k \in [T]} p_{k,i} + \frac{d_{i_2}}{\epsilon^2} \right) + L(T, \delta/\log T) \right) , \tag{15}$$

where the second inequality holds with probability at least $1 - \delta$ by Lemma 22 and the final inequality holds by the $d_i$ test. We now bound $\sum_{k \in [T]} p_{k,i}$ as

$$\sum_{k \in [T]} p_{k,i} \leq T(1 \wedge d_{i_2}^{-(\gamma+1)} d_{i_2^\star}^{\gamma+1}) \leq T d_{i_2}^{-(\gamma+1)} (8d(S_{T,n}(h), \delta))^{\gamma+1} \wedge T$$

where we used that by Lemma 11 learner $i^\star$ never gets eliminated in the considered event. $\qquad \square$

### A.2.2 Regret Analysis

To bound the overall cumulative regret of Algorithm 2, we decompose the rounds $[T]$ into the following three disjoint sets of rounds

$$[T] = \mathcal{R}_{i^\star} \dot\cup \, \mathcal{U}_{i^\star} \dot\cup \, \mathcal{O}_{i^\star}, \tag{16}$$

where

- $\mathcal{R}_{i^\star} = \{t \in [T] \colon I_{t,i^\star} = 1\}$ are the rounds where $i^\star$ requests a label,
- $\mathcal{U}_{i^\star} = \{t \in [T] \colon I_{t,i^\star} = 0, I_{t,i_t} = 0\}$ are the rounds where $i^\star$ does not request the label and the label was not observed,
- $\mathcal{O}_{i^\star} = \{t \in [T] \colon I_{t,i^\star} = 0, I_{t,i_t} = 1\}$ are the rounds where $i^\star$ does not request the label and the label was observed.

In the following three lemmas, we bound the regret in these sets of rounds separately.

**Lemma 13** (Regret in rounds where $i^\star$ requests). *In event $\bigcap_{i=\langle i_1, i_2 \rangle \in \mathcal{M}_1 \colon i_1 \geq i_1^\star} \mathcal{E}_i$, the regret in rounds where $i^\star = \langle i_1^\star, i_2^\star \rangle$ would request the label is bounded with probability at least $1 - \delta$ for all $\epsilon \in (0, 1/2)$ as*

$$R(\mathcal{R}_{i^\star}) = O\left( \frac{M}{\epsilon} 2^{\gamma+1} d(S_{i_1^\star}, \delta)^{\gamma+2} + \frac{M}{\epsilon} 2^{\gamma+1} d(S_{i_1^\star}, \delta)^{\gamma+1} L(T, \delta) + \epsilon T_\epsilon \right) . \tag{17}$$

*Proof.* In any round, the largest instantaneous regret possible is $2|h(x_{t,1}) - 1/2| = 2|h(x_{t,-1}) - 1/2| = 2|\Delta_{t,i^\star}|$, no matter whether the prediction of $i^\star$ was followed or not. Thus, the regret in rounds $\mathcal{R}_{i^\star}$ can be bounded as

$$R(\mathcal{R}_{i^\star}) \leq 2 \sum_{t \in \mathcal{R}_{i^\star}} |\Delta_{t,i^\star}| = 2 \sum_{t \in \mathcal{R}_{i^\star}} \mathbb{1}\{|\Delta_{t,i^\star}| > \epsilon\} |\Delta_{t,i^\star}| + 2\epsilon |\mathcal{R}_{i^\star}^{\epsilon}|,$$

for any $\epsilon \in (0, 1/2)$ where $\mathcal{R}_{i^\star}^{\epsilon} = \{t \in \mathcal{R}_{i^\star} \colon |\Delta_t| \leq \epsilon\}$.

On rounds $\mathcal{R}_{i^\star}$, learner $i^\star$ wants to query the label which means $\widehat{\Delta}_{t,i^\star} \leq B_{t,i^\star}$. Moreover in $\mathcal{E}_{i^\star}$, the conditions $0 \leq \widehat{\Delta}_{t,i^\star} - \widehat{\Delta}_{t,i^\star} \leq B_{t,i^\star}$ and $0 \leq \widehat{\Delta}_{t,i^\star}$ hold. Combining both inequalities gives $|\Delta_{t,i^\star}| \leq B_{t,i^\star}$ and we can further bound the display above as

$$R(\mathcal{R}_{i^\star}) \leq \sum_{t \in \mathcal{R}_{i^\star}} \mathbb{1}\{|\Delta_{t,i^\star}| > \epsilon\}(1 \wedge 2B_{t,i^\star}) + 2\epsilon|\mathcal{R}_{i^\star}^\epsilon|$$

$$\leq \sum_{t \in \mathcal{R}_{i^\star}} \mathbb{1}\{|\Delta_{t,i^\star}| > \epsilon\}\left(1 \wedge \frac{2B_{t,i^\star}^2}{\epsilon}\right) + 2\epsilon|\mathcal{R}_{i^\star}^\epsilon|$$

$$\leq \frac{2}{\epsilon} \sum_{t \in \mathcal{R}_{i^\star}} \left(\frac{\epsilon}{2} \wedge B_{t,i^\star}^2\right) + 2\epsilon|\mathcal{R}_{i^\star}^\epsilon| .$$

To bound the remaining sum, we appeal to the randomized potential lemma in Lemma 25. We denote $\underline{p}^\star = \min_{k \in [T]} p_{k,i^\star}$ the smallest probability of $i^\star$ in any round. Then Lemma 25 gives with probability at least $1 - \delta$

$$\sum_{t \in \mathcal{R}_{i^\star}} \left(\frac{\epsilon}{2} \wedge B_{t,i^\star}^2\right) \leq \sum_{t \in \mathcal{R}_{i^\star}} \left(\frac{1}{4} \wedge B_{t,i^\star}^2\right) \leq 4\gamma_{T,i^\star}^2 \sum_{t \in \mathcal{R}_{i^\star}} \left(\frac{1}{16\gamma_{T,i^\star}^2} \wedge \|\phi(x_{t,a_{t,i^\star}})\|_{Z_{t-1,i^\star}^{-1}}^2\right)$$

$$\leq 4\gamma_{T,i^\star}^2 \left(1 + \frac{3}{16\underline{p}^\star \gamma_{T,i^\star}^2} L(T,\delta)\right) + \frac{8\gamma_{T,i^\star}^2}{\underline{p}^\star}\left(1 + \frac{1}{16\gamma_{T,i^\star}^2}\right) \log \det Z_{T,i^\star}$$

$$\leq \frac{12\gamma_{T,i^\star}^2 + \frac{1}{2}}{\underline{p}^\star} \log \det Z_{T,i^\star} + \frac{3}{4\underline{p}^\star} L(T,\delta) ,$$

because $\gamma_{t,i^\star}$ is non-decreasing in $T$. Plugging this back into the previous display yields

$$R(\mathcal{R}_{i^\star}) \leq 24 \frac{\gamma_{T,i^\star}^2 + \frac{1}{24}}{\epsilon \underline{p}^\star} \log \det Z_{T,i^\star} + \frac{3}{2\epsilon \underline{p}^\star} L(T,\delta) + 2\epsilon|\mathcal{R}_{i^\star}^\epsilon|$$

$$\leq 48 \frac{d(S_{i_1^\star}, \delta)}{\epsilon \underline{p}^\star} + \frac{3}{2\epsilon \underline{p}^\star} L(T,\delta) + 2\epsilon T_\epsilon .$$

Now, Lemma 11 ensures that $i^\star$ never gets eliminated in the considered event. Therefore

$$\frac{1}{\underline{p}^\star} \leq \frac{\sum_{i \in \mathcal{M}_1} d_{i_2}^{-(\gamma+1)}}{d_{i_2^\star}^{-(\gamma+1)}} = d_{i_2^\star}^{\gamma+1} M \leq M(2d(S_{i_1^\star}, \delta))^{\gamma+1} ,$$

where the last inequality follows from Eq. (13). Plugging this bound back into the previous display yields

$$R(\mathcal{R}_{i^\star}) \leq \frac{48M}{\epsilon} 2^{\gamma+1} d(S_{i_1^\star}, \delta)^{\gamma+2} + \frac{3M}{2\epsilon} 2^{\gamma+1} d(S_{i_1^\star}, \delta)^{\gamma+1} L(T,\delta) + 2\epsilon T_\epsilon ,$$

as claimed. $\qquad\square$

**Lemma 14** (Regret in unobserved rounds where $i^\star$ does not request). *In event $\mathcal{E}_{i^\star}$,*

$$R(\mathcal{U}_{i^\star}) \leq M . \tag{18}$$

*Proof.* If $i^\star$ is not requesting the label then $i^\star$ predicts the label as $a_t^*$. From the disagreement test $i_t$ will predict the same label as $i^\star$ so there should be no regret, except when a learner gets eliminated. Since there are at most $M$ learners and the regret per round is at most 1, the total regret on rounds $\mathcal{U}_{i^\star}$ can at most be $M$. $\qquad\square$

**Lemma 15** (Regret in observed rounds where $i^\star$ does not request). *In event $\bigcap_{i=\langle i_1,i_2\rangle \in \mathcal{M}_1 : i_1 \geq i_1^\star} \mathcal{E}_i$, the regret in rounds where $i^\star$ does not request the label, but the label was still observed is bounded as*

$$R(\mathcal{O}_{i^\star})$$

$$= O\left(\sum_{i=\langle i_1,i_2\rangle \in \mathcal{M}_1} \inf_{\epsilon \in (0,1/2)} \left(\frac{d_{i_2}}{\epsilon} + T\left(\frac{\epsilon d(S_{T,n}(h), \delta)}{d_{i_2}}\right)^{\gamma+1} + \frac{L(T,\delta)}{\epsilon}\right) + ML(T, \delta/\log T)\right) .$$

*Proof.* Note that we can decompose the regret in those rounds as

$$R(\mathcal{O}_{i^\star}) = \sum_{i \neq i_*} R(\mathcal{V}_{T,i,i^\star})$$

since no regret occurs if the played action agrees with the action proposed by $i^\star$ which did not request a label and in $\mathcal{E}_{i^\star}$ does not incur any regret in such rounds. We bound $R(\mathcal{V}_{T,i,i^\star})$ by using the fact that in all but at most one of those rounds both the observed regret test and the $d_i$ test did not trigger. This gives

$$\sum_{k \in \mathcal{V}_{T,i,i^\star}} (\mathbb{1}\{a_{k,i} \neq y_k\} - \mathbb{1}\{a_{k,i^\star} \neq y_k\}) \leq \sum_{k \in \mathcal{V}_{T,i,i^\star}} 1 \wedge B_{k,i} + 1.45\sqrt{|\mathcal{V}_{T,i,i^\star}|L(|\mathcal{V}_{T,i,i^\star}|,\delta)} + 1 \,.$$

We now apply the concentration argument in Lemma 21 to bound the LHS from below as

$$\sum_{k \in \mathcal{V}_{T,i,i^\star}} (\mathbb{1}\{a_{k,i} \neq y_k\} - \mathbb{1}\{a_{k,i^\star} \neq y_k\})$$

$$\geq \sum_{k \in \mathcal{V}_{T,i,i^\star}} (h(x_{k,a_{k,i^\star}}) - h(x_{k,a_{k,i}})) - 0.72\sqrt{|\mathcal{V}_{T,i,i^\star}|L(|\mathcal{V}_{T,i,i^\star}|,\delta)}$$

$$= \sum_{k \in \mathcal{V}_{T,i,i^\star}} (h(x_{k,a_k^\star}) - h(x_{k,a_{k,i}})) - 0.72\sqrt{|\mathcal{V}_{T,i,i^\star}|L(|\mathcal{V}_{T,i,i^\star}|,\delta)}$$

$$= R(\mathcal{V}_{T,i,i^\star}) - 0.72\sqrt{|\mathcal{V}_{T,i,i^\star}|L(|\mathcal{V}_{T,i,i^\star}|,\delta)} \,,$$

where $a_k^\star$ is the optimal prediction in round $k$. Combining the previous two displays allows us to bound the regret from above for any $\epsilon \in (0,1/2)$ as

$$R(\mathcal{V}_{T,i,i^\star}) \leq \sum_{k \in \mathcal{V}_{T,i,i^\star}} (1 \wedge B_{k,i}) + 3\sqrt{|\mathcal{V}_{T,i,i^\star}|L(T,\delta)} + 1$$

$$\leq \sum_{k \in \mathcal{V}_{T,i,i^\star}} (1 \wedge I_{k,i}B_{k,i})\mathbb{1}\{B_{k,i} \geq \epsilon\} + \frac{5}{2}\epsilon|\mathcal{V}_{T,i,i^\star}| + \frac{3}{2}\frac{L(T,\delta)}{\epsilon} + 1$$

$$\leq \frac{1}{\epsilon}\sum_{k \in \mathcal{V}_{T,i,i^\star}} (\epsilon \wedge I_{k,i}B_{k,i}^2) + \frac{5}{2}\epsilon|\mathcal{V}_{T,i,i^\star}| + \frac{3}{2}\frac{L(T,\delta)}{\epsilon} + 1$$

$$\leq 8\frac{d_i}{\epsilon} + \frac{5}{2}\epsilon|\mathcal{V}_{T,i,i^\star}| + \frac{3}{2}\frac{L(T,\delta)}{\epsilon} + 1 \,,$$

where the last inequality applies the condition of the $d_i$ test. Since $\mathcal{V}_{T,i,i^\star}$ can only contain rounds where $i$ was chosen and requested a label, we can apply the label complexity bound from Eq. (15) (with $\sum_{k \in [T]} p_{k,i}$ therein upper bounded as explained just afterwards) which gives

$$|\mathcal{V}_{T,i,i^\star}| = O\left(\inf_{\epsilon \in (0,1/2)} \left(\epsilon^\gamma T \left(\frac{d(S_{T,n}(h),\delta)}{d_{i_2}}\right)^{\gamma+1} + \frac{d_{i_2}}{\epsilon^2}\right) + L(T,\delta/\log T)\right) \,, \qquad (19)$$

and plugging this back into the previous bound yields, for any $i = \langle i_1, i_2 \rangle$,

$$R(\mathcal{V}_{T,i,i^\star}) = O\left(\frac{d_{i_2}}{\epsilon} + T\left(\frac{\epsilon\, d(S_{T,n}(h),\delta)}{d_{i_2}}\right)^{\gamma+1} + \frac{L(T,\delta)}{\epsilon} + L(T,\delta/\log T)\right) \,.$$

Summing over $i \neq i^*$ gives the claimed result. $\qquad\square$

### A.2.3 Putting it all together

Putting together the above results gives rise to the following guarantee on the regret and the label complexity of Algorithm 2, presented in the main paper as Theorem 2.

**Theorem 4.** *Let Algorithm 2 be run with parameters $\delta$, $\gamma \leq \alpha$ with a pool of base learners $\mathcal{M}_1$ of size $M$ on an i.i.d. sample $(x_1, y_1), \ldots, (x_T, y_T) \sim \mathcal{D}$, where the marginal distribution $\mathcal{D}_{\mathcal{X}}$ fulfills the low-noise condition with exponent $\alpha \geq 0$ w.r.t. a function $h$ that satisfies (1) and having complexity $S_{T,n}(h)$. Let also $\mathcal{M}_1$ contain at least one base learner $i$ such that $\sqrt{2}S_{T,n}(h) \leq S_i \leq 2\sqrt{2}S_{T,n}(h)$ and $d_i = \Theta(L_H(L_H + \log(M/\delta) + S_{T,n}^2(h)))$, where $L_H = \log \det(I + H)$, being $H$ the NTK matrix of depth $n$ over the set of points $\{x_{t,a}\}_{t=1,\ldots,T, a=\pm 1}$. Also assume $m \geq CT^4 \log(2Tn/\delta)n^6 \left(T^2 \vee 1/\lambda_0^4\right)$ where $C$ is the constant in Lemma 1 and Lemma 2. Then with probability at least $1 - \delta$ the cumulative regret $R_T$ and the total number of queries $N_T$ are simultaneously upper bounded as follows:*

$$R_T = O\left(M\left(L_H\left(L_H + \log(M/\delta) + S_{T,n}^2(h)\right)\right)^{\gamma+1} T^{\frac{1}{\gamma+2}} + M L(T, \delta)\right)$$

$$N_T = O\left(M\left(L_H\left(L_H + \log(M/\delta) + S_{T,n}^2(h)\right)\right)^{\frac{\gamma}{\gamma+2}} T^{\frac{2}{\gamma+2}} + M L(T, \delta)\right),$$

*where $L(T, \delta)$ is the logarithmic term defined at the beginning of Algorithm 2's pseudocode.*

*Proof.* Using the decomposition in Eq. (16) combined with Lemmas 13, 14, and 15 we see that the regret of Algorithm 2 can be bounded as

$$R(T) \leq R(\mathcal{R}_{i_\star}) + R(\mathcal{U}_{i_\star}) + R(\mathcal{O}_{i_\star})$$

$$= O\left(\frac{M}{\epsilon} 2^{\gamma+1} d(S_{i_1^\star}, \delta)^{\gamma+2} + \frac{M}{\epsilon} 2^{\gamma+1} d(S_{i_1^\star}, \delta)^{\gamma+1} L(T, \delta) + \epsilon T_\epsilon \right.$$

$$\left. + \sum_{i=\langle i_1, i_2 \rangle \in \mathcal{M}_1} \inf_{\epsilon \in (0, 1/2)} \left(\frac{d_{i_2}}{\epsilon} + T\left(\frac{\epsilon d(S_{T,n}(h), \delta)}{d_{i_2}}\right)^{\gamma+1} + \frac{L(T, \delta)}{\epsilon}\right) + ML(T, \delta/\log T)\right).$$

We first bound term $T_\epsilon$ through Lemma 23 (Appendix A.4). This gives, with probability at least $1 - \delta$,

$$T_\epsilon = O\left(T\epsilon^\gamma + \log\frac{\log T}{\delta}\right),$$

simultaneously over $\epsilon$. Plugging back into the above, collecting terms and resorting to a big-oh notation that disregards multiplicative constants independent of $T$, $M$, $1/\delta$ yields

$$R(T) = O\left(\frac{M}{\epsilon}\left(d(S_{T,n}(h), \delta)^{\gamma+2} + d(S_{T,n}(h), \delta)^{\gamma+1} L(T, \delta)\right) + \epsilon^{\gamma+1} T + ML(T, \delta/\log T)\right.$$

$$\tag{20}$$

$$\left. + \sum_{i=\langle i_1, i_2 \rangle \in \mathcal{M}_1} \inf_{\epsilon \in (0, 1/2)} \left(\frac{d_{i_2}}{\epsilon} + T\left(\frac{\epsilon d(S_{T,n}(h), \delta)}{d_{i_2}}\right)^{\gamma+1} + \frac{L(T, \delta)}{\epsilon}\right)\right), \quad (21)$$

holding simultaneously for all $\epsilon \in (0, 1/2)$.

Now, the sum of the first two terms in the RHS (that is, Eq. (20)) is minimized by selecting $\epsilon$ of the form

$$\epsilon = \left(M\left(\frac{d(S_{T,n}(h), \delta)^{\gamma+2} + d(S_{T,n}(h), \delta)^{\gamma+1} L(T, \delta)}{T}\right)\right)^{\frac{1}{\gamma+2}}$$

which, plugged back into (20) gives

$$(20) = O\left(\left(M\left(d(S_{T,n}(h), \delta)^{\gamma+2} + d(S_{T,n}(h), \delta)^{\gamma+1} L(T, \delta)\right)\right)^{\frac{\gamma+1}{\gamma+2}} T^{\frac{1}{\gamma+2}} + ML(T, \delta/\log T)\right)$$

$$= O\left(M d(S_{T,n}(h), \delta)^{\gamma+1} T^{\frac{1}{\gamma+2}} L(T, \delta/\log T)\right).$$

Notice that $\epsilon$ is constrained to lie in $(0, 1/2)$. If that is not the case with the above choice of $\epsilon$, our bound delivers vacuous regret guarantees.

As for the sum in (21), each term in the sum is individually minimized by an $\epsilon$ of the form

$$\epsilon = \left( \frac{(d_{i_2} + L(T, \delta)) \cdot d_{i_2}^{\gamma+1}}{T \cdot d(S_{T,n}(h), \delta)^{\gamma+1}} \right)^{\frac{1}{\gamma+2}}.$$

Notice that the above value of $\epsilon$ lies in the range $(0, \frac{1}{2})$ provided $d_{i_2} = o(T^{\frac{1}{\gamma+2}})$. Hence we simply assume that our model selection algorithm is performed over base learners with $d_{i_2}$ bounded as above. In fact, if $d(S_{T,n}(h), \delta)$ exceeds this range then our bounds become vacuous.

Next, substituting the value of $\epsilon$ obtained above we get that Eq. (21) can be bounded as

$$(21) = O\left( M d(S_{T,n}(h), \delta)^{\frac{\gamma+1}{\gamma+2}} T^{\frac{1}{\gamma+2}} \right).$$

Combining the bounds on Eq. (20) and Eq. (21) we get the claimed bound on the regret $R_T$.

Next, we bound the label complexity of the our model selection procedure. From Lemma 12 we have that the label complexity can be bounded by

$$N_T = O\left( \sum_{i=\langle i_1, i_2 \rangle \in \mathcal{M}_1} \left( \frac{d_{i_2}}{\epsilon^2} + \epsilon^\gamma T \left( 1 \wedge \frac{d(S_{T,n}(h), \delta)}{d_{i_2}} \right)^{\gamma+1} \right) + ML(T, \delta/\log T) \right) . \quad (22)$$

Next consider a term in the summation in Eq. (22) with $d_{i_2} \geq d(S_{T,n}(h), \delta)$. The following value of $\epsilon$ minimizes the term:

$$\epsilon = \left( \frac{d_{i_2}}{T^{\frac{1}{\gamma+2}}} d(S_{T,n}(h), \delta)^{-\frac{\gamma+1}{\gamma+2}} \right).$$

Again we notice that this is a valid range of $\epsilon$ provided that $d_{i_2} = o(T^{\frac{1}{\gamma+2}})$. Substituting back into Eq. (22) we obtain that the label complexity incurred due to such terms (denoted by $N_1(T)$) is bounded as

$$N_1(T) = O\left( M \frac{T^{\frac{2}{\gamma+2}} d(S_{T,n}(h), \delta)^{\frac{2(\gamma+1)}{\gamma+2}}}{d_{i_2}} + ML(T, \delta/\log T) \right)$$

$$= O\left( M T^{\frac{2}{\gamma+2}} d(S_{T,n}(h), \delta)^{\frac{\gamma}{\gamma+2}} + ML(T, \delta/\log T) \right). \quad (23)$$

Finally, consider a term in the summation in Eq. (22) with $d_{i_2} < d(S_{T,n}(h), \delta)$. Then the value of $\epsilon$ that minimizes the term equals

$$\epsilon = \left( \frac{d_{i_2}}{T} \right)^{\frac{1}{\gamma+2}}.$$

Substituting back into Eq. (22), we get that the label complexity incurred by such terms (denoted by $N_2(T)$) is bounded by

$$N_2(T) = O\left( M T^{\frac{2}{\gamma+2}} d(S_{T,n}(h), \delta)^{\frac{\gamma}{\gamma+2}} + ML(T, \delta/\log T) \right). \quad (24)$$

Noting that $N_T = N_1(T) + N_2(T)$, we get the claimed bound on the label complexity of the algorithm. $\qquad \square$

### A.3 Extension to non-Frozen NTK

Following [45], in order to avoid computing $f(x, \theta_0)$ for each input $x$, we replace each vector $x_{t,a} \in \mathbb{R}^{2d}$ by $[x_{t,a}, x_{t,a}]/\sqrt{2} \in \mathbb{R}^{4d}$, matrix $W_l$ by $\begin{pmatrix} W_l & 0 \\ 0 & W_l \end{pmatrix} \in \mathbb{R}^{4d \times 4d}$, for $l = 1, \ldots, n-1$, and $W_n$ by $\left( W_n^\top, -W_n^\top \right)^\top \in \mathbb{R}^{2d}$. This ensures that the initial output of neural network $f(x, \theta_0)$ is always 0 for any $x$.

---

**Algorithm 3:** NTK Selective Sampler.

---

**Input:** Confidence level $\delta$, complexity parameter $S$, network width $m$ and depth $n$, number of rounds $T$, step size $\eta$, number of gradient descent steps $J$.

**Initialization:**

- Generate each entry of $W_k$ independently from $\mathcal{N}(0, 4/m)$, for $k \in [n-1]$, and each entry of $W_n$ independently from $\mathcal{N}(0, 2/m)$;

- Define $\phi_t(x) = g(x; \theta_{t-1})/\sqrt{m}$, where $\theta_{t-1} = \langle W_1, \ldots, W_n \rangle \in \mathbb{R}^p$ is the weight vector of the neural network so generated at round $t-1$;

- Set $Z_0 = I \in \mathbb{R}^{p \times p}$.

for $t = 1, 2, \ldots, T$

    Observe instance $x_t \in \mathcal{X}$ and build $x_{t,a} \in \mathcal{X}^2$, for $a \in \mathcal{Y} = \{-1, +1\}$

    Set $\mathcal{C}_{t-1} = \{\theta : \|\theta - \theta_{t-1}\|_{Z_{t-1}} \leq \frac{\gamma_{t-1}}{\sqrt{m}}\}$, with $\gamma_{t-1} = 3(\sqrt{\log \det Z_{t-1} + 3\log(1/\delta)} + S)$

    Set

$$U_{t,a} = f(x_{t,a}, \theta_{t-1}) + \gamma_{t-1}\|\phi_{t-1}(x_{t,a})\|_{Z_{t-1}^{-1}} + \frac{1}{\sqrt{T}}$$

    Predict $a_t = \arg\max_{a \in \mathcal{Y}} U_{t,a}$

    Set $I_t = \mathbb{1}\{|U_{t,a_t} - 1/2| \leq B_t\} \in \{0, 1\}$    with    $B_t = 2\gamma_{t-1}\|\phi_{t-1}(x_{t,a_t})\|_{Z_{t-1}^{-1}} + \frac{2}{\sqrt{T}}$

    if $I_t = 1$

        Query $y_t \in \mathcal{Y}$, and set loss $\ell_t = \ell(a_t, y_t)$

        Update

$$Z_t = Z_{t-1} + \phi_t(x_{t,a_t})\phi_t(x_{t,a_t})^\top$$

$$\theta_t = \text{TrainNN}\left(\eta, J, m, \{x_{s,a_s} \mid s \in [t], I_s = 1\}, \{\ell_s \mid s \in [t], I_s = 1\}, \theta_0\right)$$

    else    $Z_t = Z_{t-1}, \ \theta_t = \theta_{t-1}, \ \gamma_t = \gamma_{t-1}, \ \mathcal{C}_t = \mathcal{C}_{t-1}$.

---

**Algorithm 4:** TrainNN($\eta, J, m, \{x_i\}_{i=1}^l, \{\ell_i\}_{i=1}^l, \theta^{(0)}$)

---

**Input:** Step size $\eta$, number of gradient descent steps $J$, network width $m$, contexts $\{x_i\}_{i=1}^l$, loss values $\{\ell_i\}_{i=1}^l$, initial weight $\theta^{(0)}$.

Set $\mathcal{L}(\theta) = \sum_{i=1}^l (f(x_i, \theta) - 1 + \ell_i)^2/2 + m\|\theta - \theta^{(0)}\|_2^2$.

for $j = 0, \ldots, J-1$

    $\theta^{(j+1)} = \theta^{(j)} - \eta\nabla\mathcal{L}(\theta^{(j)})$

**Return** $\theta^{(J)}$

---

### A.3.1   Non-Frozen NTK Base Learner

The pseudocode for the base learner in the non-frozen case is contained in Algorithm 3. Unlike Algorithm 1, Algorithm 3 updates $\theta_t$ using gradient descent. The update of $\theta_t$ is handled by the pseudocode in Algorithm 4.

Note that both Algorithm 1 and Algorithm 3 determine the confidence ellipsoid $\mathcal{C}_t$ by updating $\theta_t$, $\gamma_t$ and $Z_t$. To tell apart the two learners, we use $\bar{\gamma}_t$, $\bar{Z}_t$ and $\bar{\theta}_t$ to denote the ellipsoid parameters for Algorithm 1. We make use of a few relevant lemmas from [45] and its references therein stating that in the over-parametrized regime, i.e., when $m = \text{poly}(T, n, \lambda_0^{-1}, S^{-1}, \log(1/\delta))$, the gradient descent update does not leave $\theta_t$ and $Z_t$ too far from the corresponding $\bar{\theta}_t$ and $\bar{Z}_t$. Moreover, the neural network $f$ is close to its first order approximation. The interested reader is referred to Lemmas B.2 through B.6 of [45]. Combining these results with the analysis in Section A.1 we bound the label complexity and regret for Algorithm 3.

The below proofs are mainly sketched, since they follow from a combination of the arguments in Section A.1 and some technical lemmas in [45].

We re-define here $\mathcal{E}_0$ to be the event where (4) and (5) hold along with all the bounds in the well-approximation lemmas of [45] (Lemmas B.2 throug B.6). From [45], there exists a constant $C$ such that if
$$m \geq CT^{19}n^{27}(\log m)^3$$
then $\mathbb{P}(\mathcal{E}_0) \geq 1 - \delta$. Event $\mathcal{E}$ is defined as in Eq. (7) with this specific event $\mathcal{E}_0$ therein.

We give a new version of Lemma 3 below, which implies that event $\mathcal{E}$ still holds with high probability for Algorithm 3, with a specific learning rate $\eta$, number of gradient descent steps $J$ and network width $m$.

**Lemma 16.** *There exist positive constants $\bar{C}_1, \bar{C}_2$ such that if*

$$\eta = \frac{\bar{C}_1}{2mnT} , \qquad J = \frac{4nT}{\bar{C}_1} \log \frac{S}{CnT^{3/2}} , \qquad m \geq \bar{C}_2 T^{19}n^{27}(\log m)^3$$

*and $\sqrt{2}S_{T,n}(h) \leq S$, then under event $\mathcal{E}_0$ for any $\delta \in (0,1)$ we have with probability at least $1 - \delta$*

$$\|\theta^* - \theta_t\|_{Z_t} \leq \gamma_t/\sqrt{m}$$

*simultaneously for all $t > 0$. In other words, under event $\mathcal{E}_0$, $\theta^* \in \mathcal{C}_t$ with high probability for all $t$.*

*Proof sketch.* In Lemma 5.2 of [45], it is shown that

$$\sqrt{m}\|\theta^* - \theta_t\|_{Z_t} \leq \sqrt{1 + Cm^{-1/6}\sqrt{\log m}n^4 t^{7/6}}$$
$$\times \left( \sqrt{\log \det Z_t + Cm^{-1/6}\sqrt{\log m}n^4 t^{5/3} + 2\log(1/\delta)} + S \right)$$
$$+ Cn\left( (1 - \eta m)^{J/2}t^{3/2} + Cm^{-1/6}\sqrt{\log m}n^{7/2}t^{19/6} \right)$$

for some constant $C$ under event $\mathcal{E}_0$ and the assumption that $\sqrt{2}S_{T,n}(h) \leq S$. Setting $\eta = \frac{\bar{C}_1}{2mnT}$ and $J = \frac{4nT}{\bar{C}_1} \log \frac{S}{CnT^{3/2}}$ allows us to bound $Cn(1 - \eta m)^{J/2}T^{3/2}$ by $S$. Lastly, since $m$ satisfies

$$\frac{C^2\sqrt{\log m}\, n^{9/2}T^{19/6}}{m^{1/6}} \leq 1 ,$$

we have

$$\sqrt{m}\|\theta^* - \theta_t\|_{Z_t} \leq \sqrt{2}\left( \sqrt{\log \det Z_t + 1 + 2\log(1/\delta)} + S \right) + S + 1$$
$$\leq 3\left( \sqrt{\log \det Z_t + 3\log(1/\delta)} + S \right) ,$$

as claimed. $\qquad\square$

We next show the properties of $\widehat{\Delta}_t$ and $\Delta_t$, which is a new version of Lemma 6 for the non-frozen case.

**Lemma 17.** *Assume $m \geq poly(T, n, \lambda_0^{-1}, S, \log(1/\delta))$ and $\sqrt{2}S_{T,n}(h) \leq S$. Then under event $\mathcal{E}$ we have $0 \leq \widehat{\Delta}_t - \Delta_t \leq B_t$ and $0 \leq \widehat{\Delta}_t$, where $B_t$ is the querying threshold in Algorithm 3, i.e.,*

$$B_t = 2\gamma_{t-1}\|\phi_t(x_{t,a_t})\|_{Z_{t-1}^{-1}} + \frac{2}{\sqrt{T}} .$$

*Proof.* Denote

$$\tilde{U}_{t,a} = \max_{\theta \in \mathcal{C}_{t-1}} \langle g(x_{t,a}; \theta_{t-1}), \theta - \theta_0 \rangle = \langle g(x_{t,a}; \theta_{t-1}), \theta_{t-1} - \theta_0 \rangle + \gamma_{t-1}\|\phi_t(x_{t,a})\|_{Z_{t-1}^{-1}} .$$

We decompose

$$\widehat{\Delta}_t - \Delta_t = (U_{t,a} - \tilde{U}_{t,a}) + (\tilde{U}_{t,a} - h(x_{t,a})) =: A_1 + A_2 .$$

For $A_1$, by definition of $U_{t,a}$ in Algorithm 3 we have

$$U_{t,a} - \tilde{U}_{t,a} = f(x_{t,a}; \theta_{t-1}) - \langle g(x_{t,a}; \theta_{t-1}), \theta_{t-1} - \theta_0 \rangle + \frac{1}{\sqrt{T}} .$$

Under event $\mathcal{E}$, the bound in Lemma B.4 of [45] holds. That is, there is a constant $C_2$ such that

$$|f(x_{t,a}; \theta_{t-1}) - \langle g(x_{t,a}; \theta_{t-1}), \theta_{t-1} - \theta_0 \rangle|$$
$$= |f(x_{t,a}; \theta_{t-1}) - f(x_{t,a}; \theta_0) - \langle g(x_{t,a}; \theta_{t-1}), \theta_{t-1} - \theta_0 \rangle|$$
$$\leq C_2 m^{-1/6} \sqrt{\log m} n^3 t^{2/3} .$$

Setting $m$ so large as to satisfy $C_2 m^{-1/6} \sqrt{\log m} n^3 T^{2/3} \leq \frac{1}{2\sqrt{T}}$ gives us

$$\frac{1}{2\sqrt{T}} \leq A_1 \leq \frac{3}{2\sqrt{T}} .$$

To estimate $A_2$ we decompose it further as

$$A_2 = \left( \tilde{U}_{t,a} - \langle g(x_{t,a}; \theta_{t-1}), \theta^\star - \theta_0 \rangle \right) + \left( \langle g(x_{t,a}; \theta_{t-1}), \theta^\star - \theta_0 \rangle - \langle g(x_{t,a}; \theta_0), \theta^\star - \theta_0 \rangle \right)$$
$$=: A_3 + A_4 .$$

Following the argument in Lemma 6 we can show the inequality $0 \leq A_3 \leq 2\gamma_{t-1} \|\phi_t(x_{t,a_t})\|_{Z_{t-1}^{-1}}$ under event $\mathcal{E}$. By Cauchy-Schwartz inequality $|A_4| \leq \|g(x_{t,a}; \theta_{t-1}) - g(x_{t,a}; \theta_0)\|_2 \|\theta^\star - \theta_0\|_2$. Using the assumption that the bounds in Lemmas B.5 and B.6 in [45] hold and $\sqrt{2} S_{T,n}(h) \leq S$, there exists a constant $C_1$ such that

$$|A_4| \leq \|g(x_{t,a}; \theta_{t-1}) - g(x_{t,a}; \theta_0)\|_2 \|\theta^\star - \theta_0\|_2 \leq C_1 S m^{-1/6} \sqrt{\log m} n^{7/2} t^{1/6} .$$

Setting $m$ large enough to satisfy $C_1 S m^{-1/6} \sqrt{\log m} n^{7/2} T^{1/6} \leq \frac{1}{2\sqrt{T}}$ gives us

$$-\frac{1}{2\sqrt{T}} \leq A_2 \leq 2\gamma_{t-1} \|\phi_t(x_{t,a_t})\|_{Z_{t-1}^{-1}} + \frac{1}{2\sqrt{T}} .$$

Combining the bound for $A_1$ and $A_2$ we obtain

$$0 \leq \widehat{\Delta}_t - \Delta_t \leq B_t ,$$

which proves the first part of the claim.

Next, since $U_{t,a} - h(x_{t,a}) \geq 0$ for $a \in \mathcal{Y}$, we also have

$$U_{t,1} + U_{t,-1} \geq h(x_{t,1}) + h(x_{t,-1}) = 1$$

which, by definition of $a_t$, gives $U_{t,a_t} \geq \frac{1}{2}$, i.e., $\widehat{\Delta}_t \geq 0$. This concludes the proof. $\qquad \square$

As a consequence of the above lemma, like in the frozen case, on rounds where Algorithm 3 does not issue a query, we are confident that prediction $a_t$ suffers no regret.

Before bounding the label complexity and regret, we give the following lemma which is the non-frozen counterpart to Lemma 5 in Section A.1. The proof follows from very similar arguments, and is therefore omitted.

**Lemma 18.** *Let $\eta$, $J$ and $m$ be as in Lemma 16 and $\sqrt{2} S_{T,n}(h) \leq S$. Then for any $b > 0$ we have*

$$\sum_{t=1}^{T} b \wedge I_t B_t^2 = O\left( \left( \log \det Z_T + \log(1/\delta) + S^2 + b \right) \log \det Z_T \right) . \qquad (25)$$

Combining the above lemmas we can bound the label complexity and regret similar to Section A.1.

**Lemma 19.** *Let $\eta$, $J$ be as in Lemma 16, $m \geq poly(T, n, \lambda_0^{-1}, S, \log(1/\delta))$, and $\sqrt{2} S_{T,n}(h) \leq S$. Then under event $\mathcal{E}$ for any $\epsilon \in (0, 1/2)$ we have*

$$N_T = O\left( T_\epsilon + \frac{1}{\epsilon^2} (\log \det Z_T + \log(1/\delta) + S^2) \log \det Z_T \right)$$
$$= O\left( T_\epsilon + \frac{1}{\epsilon^2} \left( \log \det(I + H) + \log(1/\delta) + S^2 \right) \log \det(I + H) \right) .$$

**Lemma 20.** *Let $\eta$, $J$ be as in Lemma 16, $m \geq poly(T, n, \lambda_0^{-1}, S, \log(1/\delta))$, and $\sqrt{2}S_{T,n}(h) \leq S$. Then under event $\mathcal{E}$ for any $\epsilon \in (0, 1/2)$ we have,*

$$R_T = O\left(\epsilon T_\epsilon + \frac{1}{\epsilon}\left(\log\det Z_T + \log(1/\delta) + S^2\right)\log\det Z_T\right)$$

$$= O\left(\epsilon T_\epsilon + \frac{1}{\epsilon}\left(\log\det(I + H) + \log(1/\delta) + S^2\right)\log\det(I + H)\right).$$

The rest of the analysis follows from the same argument that relies on Lemma 23 (Appendix A.4) allowing one to replace $T_\epsilon$ by $O\left(T\epsilon^\alpha + O\left(\log\frac{\log T}{\delta}\right)\right)$, and culminating into a statement very similar to Theorem 1.

### A.3.2 Model Selection for Non-Frozen NTK Base Learners

The pseudocode for the model selection algorithm applied to the case where the base learners are of the form of Algorithm 3 instead of Algorithm 1 is very similar to Algorithm 2, and so is the corresponding analysis. The adaptation to non-frozen base learners simply requires to change a constant. Specifically, we replace '8' in the $d_i$ test of Algorithm 2 with '432', all the rest remains the same, provided the definition of $B_{t,i}$ (querying threshold of the $i$-th base learner) is now taken from Algorithm 3 ($B_t$ therein).

An analysis very similar to Lemma 11 shows that a well-specified learner is (with high probability) not removed from the pool $\mathcal{M}_t$, while the label complexity and the regret analyses mimic the corresponding analyses contained in Section A.2.1 and A.2.2, with inflated constants and network width $m$.

### A.4 Ancillary technical lemmas

**Lemma 21.** *Let $i, j \in \mathcal{M}_1$ be two base learners. with probability at least $1 - 2\delta$ the following concentration bound holds for all rounds $t$*

$$\left|\sum_{k \in \mathcal{V}_{t,i,j}}\left(\mathbb{1}\{a_{k,i} \neq y_k\} - \mathbb{1}\{a_{k,j} \neq y_k\} + h(x_{k,a_{k,i}}) - h(x_{k,a_{k,j}})\right)\right| \leq 0.72\sqrt{|\mathcal{V}_{t,i,j}|L(|\mathcal{V}_{t,i,j}|, \delta)}.$$

*Proof.* We write the LHS of the inequality to show as $\left|\sum_{k=1}^t Y_k\right|$ where

$$Y_k = \mathbb{1}\{k \in \mathcal{V}_{t,i,j}\}(\mathbb{1}\{a_{k,j} = y_k\} - \mathbb{1}\{a_{k,i} = y_k\} + h(x_{k,a_{k,i}}) - h(x_{k,a_{k,j}})).$$

and let $\mathbb{E}_k$ and $\text{Var}_k$ denote expectation and variance conditioned on everything before $y_k$ (including $x_k, a_{k,i}, a_{k,j}$ and $i_k$). Note that $Y_k$ is a martingale difference sequence since $\mathbb{E}_k Y_k = 0$. Further, $H_k = \mathbb{1}\{k \in \mathcal{V}_{t,i,j}\}(1 + h(x_{k,a_{k,i}}) - h(x_{k,a_{k,j}}))$ and $G_k = -\mathbb{1}\{k \in \mathcal{V}_{t,i,j}\}(-1 + h(x_{k,a_{k,i}}) - h(x_{k,a_{k,j}}))$ are predictable sequences with $-G_k \leq Y_k \leq H_k$. Thus, we can apply Lemma 27 and get that with probability at least $1 - \delta$, for all $t \in \mathbb{N}$

$$\sum_{i=1}^t Y_i \leq 1.44\sqrt{(W_t \vee m)\left(1.4\log\log\left(2\left(\frac{W_t}{m} \vee 1\right)\right) + \log\frac{5.2}{\delta}\right)}$$

$$\leq 0.72\sqrt{|\mathcal{V}_{t,i,j}|\left(1.4\log\log\left(2|\mathcal{V}_{t,i,j}|\right) + \log\frac{5.2}{\delta}\right)} = 0.72\sqrt{|\mathcal{V}_{t,i,j}|L(|\mathcal{V}_{t,i,j}|, \delta)}$$

where $W_t = |\mathcal{V}_{t,i,j}|/4$ and $m = 1/4$. We can apply the same argument to $-Y_k$ which yields the statement to show. $\square$

**Lemma 22.** *For any $i \in \mathcal{M}_1$ the number of rounds in which $i$ was played is bounded with probability at least $1 - \delta$ for all $t \in [T]$ as*

$$|\mathcal{T}_{t,i}| \leq \frac{3}{2}\sum_{k=1}^t p_{k,i} + 1.45L(t, \delta).$$

*Proof.* We can write the size of $T_{t,i}$ by its definition as $|\mathcal{T}_{t,i}| = \sum_{k=1}^{t} \mathbb{1}\{i_k = i\}$. We denote by $\mathcal{F}_k$ the $\sigma$-field induced by all observed quantities in Algorithm 2 before $i_k$ is sampled (including the set of active learners $\mathcal{M}_k$). By construction $(\mathcal{F}_t)_{t\in\mathbb{N}}$ is a filtration. Note further that $\mathbb{1}\{i_k = i\}$ conditioned on $\mathcal{F}_k$ is Bernoulli random variable with probability $p_{k,i}$. We can therefore apply Lemma 26 with $Y_k = \mathbb{1}\{i_k = i\} - p_{k,i}$, $m = p_{1,i}$ (which is a fixed quantity) and $W_t = \sum_{k=1}^{t} p_{k,i}(1 - p_{k,i}) \leq \sum_{k=1}^{t} p_{k,i}$. This gives that with probability at least $1 - \delta$

$$\sum_{k=1}^{t} \mathbb{1}\{i_k = i\} - \sum_{k=1}^{t} p_{k,i} \leq 1.44\sqrt{L(t,\delta)\sum_{k=1}^{t} p_{k,i}} + 0.41 L(t,\delta)$$

$$\leq \frac{1}{2}\sum_{k=1}^{t} p_{k,i} + 1.45 L(t,\delta).$$

Note that $W_t/p_{1,i} \leq t$ holds because the smallest non-zero probability $p_{k,i}$ is $p_{1,i}$. Rearranging terms yields the desired statement. $\square$

**Lemma 23.** *Under the low-noise assumption with exponent $\alpha \geq 0$, each of the following three bounds holds for any $i \in [M]$ with probability at least $1 - \log_2(12T)\delta$:*

$$\forall t \in [T], \epsilon \in (0, 1/2)\colon \quad |\mathcal{T}_{t,i}^{\epsilon}| \leq 3\epsilon^{\alpha}\sum_{k=1}^{t} p_{k,i} + 2L(t,\delta), \tag{26}$$

$$\forall t \in [T], \epsilon \in (0, 1/2)\colon \quad |\mathcal{T}_{t,i}^{\epsilon}| \leq 3\epsilon^{\alpha}|\mathcal{T}_{t,i}| + 2L(|\mathcal{T}_{t,i}|, \delta), \tag{27}$$

$$\epsilon \in (0, 1/2)\colon \quad T_{\epsilon} \leq 3\epsilon^{\alpha}T + 2L(T, \delta). \tag{28}$$

*Proof.* We here show the result for Eq. (26). The arguments for Eq. (27) and Eq. (28) follow analogously (by considering $\mathbb{1}\{i_k = i\}$ and 1 instead of $p_{k,i}$). To show Eq. (26), we first prove this condition for a *fixed* $\epsilon \in (0, 1/2]$: We begin by writing $T_{t,i}^{\epsilon}$ by its definition as

$$|\mathcal{T}_{t,i}^{\epsilon}| = \sum_{k=1}^{t} \mathbb{1}\{i_k = i\}\mathbb{1}\{|\Delta_k| \leq \epsilon\}.$$

We denote by $\mathcal{F}_k$ the $\sigma$-field induced by all quantities determined up to the end of round $k - 1$ in Algorithm 2 (including the set of active learners $\mathcal{M}_k$ but not $i_k$ or $x_k$). By construction $(\mathcal{F}_t)_{t\in\mathbb{N}}$ is a filtration. Conditioned on $\mathcal{F}_k$, the r.v. $\mathbb{1}\{i_k = i\}\mathbb{1}\{|\Delta_k| \leq \epsilon\}$ is a Bernoulli random variables with probability $q_k \leq p_{k,i}\epsilon^{\alpha}$, because the choice of learner and the distribution of $|\Delta_k| \leq \epsilon$ are independent in each round and by low noise condition, the latter is at most $\epsilon^{\alpha}$. We can therefore apply Lemma 26 with $Y_k = \mathbb{1}\{i_k = i\}\mathbb{1}\{|\Delta_k| \leq \epsilon\} - q_k$, $m = q_1$ and $W_t = \sum_{k=1}^{t} q_k(1 - q_k) \leq \sum_{k=1}^{t} q_k$. This gives that with probability at least $1 - \delta$

$$\sum_{k=1}^{t} \mathbb{1}\{i_k = i\}\mathbb{1}\{|\Delta_k| \leq \epsilon\} - \sum_{k=1}^{t} q_k \leq 1.44\sqrt{L(t,\delta)\sum_{k=1}^{t} q_k} + 0.41 L(t,\delta)$$

$$\leq \frac{1}{2}\sum_{k=1}^{t} q_k + 1.45 L(t,\delta),$$

where the second inequality follows from AM-GM. Rearranging terms and using $q_k \leq p_{k,i}\epsilon^{\alpha} \leq p_{k,i}$ gives for a fixed $\epsilon$

$$|\mathcal{T}_{t,i}^{\epsilon}| \leq \frac{3}{2}\epsilon^{\alpha}\sum_{k=1}^{t} p_{k,i} + 1.45 L(t,\delta). \tag{29}$$

We now consider the following set of values for $\epsilon$

$$\mathcal{K} = \left\{ \left(\frac{1}{3T}\right)^{1/\alpha} 2^{\frac{i-1}{\alpha}} : i = 1, \ldots, \left\lfloor \log_2\left(\frac{3T}{2^{\alpha-1}}\right) \right\rfloor \right\} \cup \{1/2\}.$$

and apply the argument above for all $\epsilon \in \mathcal{K}$ which gives that with probability at least $1 - \delta|\mathcal{K}| \geq 1 - \log_2(12T)\delta$, the bound in Eq. (29) holds for all $\epsilon \in \mathcal{K}$ and $t \in \mathbb{N}$ simultaneously. In this event, consider any arbitrary $\epsilon \in (0, 1/2)$ and $t \in [T]$. Then

$$|\mathcal{T}_{t,i}^\epsilon| \leq |\mathcal{T}_{t,i}^{\epsilon'}| \leq \frac{3}{2}\epsilon'^\alpha \sum_{k=1}^t p_{k,i} + 1.45L(t, \delta),$$

where $\epsilon' = \min\{x \in \mathcal{K} \colon x \geq \epsilon\}$. If $\epsilon'$ is the smallest value in $\mathcal{K}$, then $\frac{3}{2}\epsilon'^\alpha \sum_{k=1}^t p_{k,i} \leq 1/2 \leq 1/2L(t, \delta)$. Thus, the RHS is bounded as $2L(t, \delta)$ in this case. If $\epsilon'$ is not the smallest value in $\mathcal{K}$, then by construction $2\epsilon^\alpha \geq \epsilon'^\alpha$ and the RHS is bounded as $\frac{3}{2}\epsilon'^\alpha \sum_{k=1}^t p_{k,i} + 1.45L(t, \delta) \leq 3\epsilon^\alpha \sum_{k=1}^t p_{k,i} + 1.45L(t, \delta)$. Combining both cases gives the desired result for Eq. (26). $\square$

**Lemma 24** (Elliptical potential, Lemma C.2 [35]). *Let $x_1, \ldots, x_n \in \mathbb{R}^d$ and $V_t = V_0 + \sum_{i=1}^t x_i x_i^\top$ and $b > 0$ then*

$$\sum_{t=1}^n b \wedge \|x_t\|_{V_{t-1}^{-1}}^2 \leq \frac{b}{\log(b+1)} \log \frac{\det V_n}{\det V_0} \leq (1+b) \log \frac{\det V_n}{\det V_0}.$$

**Lemma 25** (Randomized elliptical potential). *Let $x_1, x_2, \cdots \in \mathbb{R}^d$ and $I_1, I_2, \cdots \in \{0, 1\}$ and $V_0 \in \mathbb{R}^{d \times d}$ be random variables so that $\mathbb{E}[I_k | x_1, I_1, \ldots, x_{k-1}, I_{k-1}, x_k, V_0] = p_k$ for all $k \in \mathbb{N}$. Further, let $V_t = V_0 + \sum_{i=1}^t I_i x_i x_i^\top$. Then*

$$\sum_{t=1}^n b \wedge \|x_t\|_{V_{t-1}^{-1}}^2 \leq 1 \vee 2.9 \frac{b}{p} \left( 1.4 \log \log (2bn \vee 2) + \log \frac{5.2}{\delta} \right) + \frac{2}{p}(1+b) \log \frac{\det V_n}{\det V_0}$$

*holds with probability at least $1 - \delta$ for all $n$ simultaneously where $p = \min_k p_k$ is the smallest probability.*

*Proof.* This proof is a slight generalization of the Lemma C.4 in [35]. We provide the full proof here for convenience: We decompose the sum of squares as

$$\sum_{t=1}^n b \wedge \|x_t\|_{V_{t-1}^{-1}}^2 \leq \frac{1}{p} \sum_{t=1}^n (bI_t \wedge \|I_t x_t\|_{V_{t-1}^{-1}}^2) + \sum_{t=1}^n \frac{1}{p_t}(p_t - I_t)(b \wedge \|x_t\|_{V_{t-1}^{-1}}^2) \qquad (30)$$

The first term can be controlled using the standard elliptical potential lemma in Lemma 24 as

$$\frac{1}{p} \sum_{t=1}^n (bI_t \wedge \|I_t x_t\|_{V_{t-1}^{-1}}^2) \leq \frac{1}{p}(1+b) \ln \frac{\det V_n}{\det V_0}.$$

For the second term, we apply an empirical variance uniform concentration bound. Let $\mathcal{F}_{i-1} = \sigma(V_0, x_1, p_1, I_1, \ldots, x_{i-1}, I_{i-1}, x_i, p_i)$ be the sigma-field up to before the $i$-th indicator. Let $Y_i = \frac{1}{p_i}(p_i - I_i) \left( \|x_i\|_{V_{i-1}^{-1}}^2 \wedge b \right)$ which is a martingale difference sequence because $\mathbb{E}[Y_i | \mathcal{F}_{i-1}] = 0$ and consider the process $S_t = \sum_{i=1}^t Y_i$ with variance process

$$W_t = \sum_{i=1}^t \mathbb{E}[Y_i^2 | \mathcal{F}_{i-1}] = \sum_{i=1}^t \frac{1}{p_i^2} \left( \|x_i\|_{V_{i-1}^{-1}}^2 \wedge b \right)^2 \mathbb{E}[(p - I_i)^2 | \mathcal{F}_{i-1}]$$

$$= \sum_{i=1}^t \frac{1 - p_i}{p_i} \left( \|x_i\|_{V_{i-1}^{-1}}^2 \wedge b \right)^2 \leq \sum_{i=1}^t \frac{b}{p_i} \left( \|x_i\|_{V_{i-1}^{-1}}^2 \wedge b \right) \leq \sum_{i=1}^t \frac{b^2}{p_i}.$$

Note that $Y_t \leq b$ and therefore, $S_t$ satisfies with variance process $W_t$ the sub-$\psi_P$ condition of [22] with constant $c = b$ (see Bennett case in Table 3 of [22]). By Lemma 26 below, the bound

$$S_t \leq 1.44 \sqrt{(W_t \vee m) \left( 1.4 \ln \ln (2(W_t/m \vee 1)) + \ln \frac{5.2}{\delta} \right)}$$

$$+ 0.41b \left( 1.4 \ln \ln (2(W_t/m \vee 1)) + \ln \frac{5.2}{\delta} \right)$$

holds for all $t \in \mathbb{N}$ with probability at least $1 - \delta$. We set $m = \frac{b}{p}$ and upper-bound the RHS further as

$$
1.44 \sqrt{\frac{b}{p} \left( 1 \vee \sum_{i=1}^{t} \left( b \wedge \|x_i\|_{V_{i-1}^{-1}}^2 \right) \right) \left( 1.4 \ln \ln \left( 2bt \vee 2 \right) + \ln \frac{5.2}{\delta} \right)}
$$

$$
+ 0.41 b \left( 1.4 \ln \ln \left( 2bt \vee 2 \right) + \ln \frac{5.2}{\delta} \right)
$$

$$
\leq \frac{1}{2} \left( 1 \vee \sum_{i=1}^{t} \left( b \wedge \|x_i\|_{V_{i-1}^{-1}}^2 \right) \right) + 1.45 \frac{b}{p} \left( 1.4 \ln \ln \left( 2bt \vee 2 \right) + \ln \frac{5.2}{\delta} \right),
$$

where the inequality is an application of the AM-GM inequality. Thus, we have shown that with probability at least $1 - \delta$, for all $n$, the second term in Eq. (30) is bounded as

$$
\frac{1}{p} \sum_{t=1}^{n} (p_t - I_t)(b \wedge \|x_t\|_{V_{t-1}^{-1}}^2) \leq \frac{1}{2} \left( 1 \vee \sum_{i=1}^{n} \left( \|x_i\|_{V_{i-1}^{-1}}^2 \wedge b \right) \right) + Z.
$$

where $Z = 1.45 \frac{b}{p} \left( 1.4 \ln \ln \left( 2bn \vee 2 \right) + \ln \frac{5.2}{\delta} \right)$. And when combining all bounds on the sum of squares term in Eq. (30), we get that either $\sum_{i=1}^{n} \left( \|x_i\|_{V_{i-1}^{-1}}^2 \wedge b \right) \leq 1$ or

$$
\sum_{i=1}^{n} \left( \|x_i\|_{V_{i-1}^{-1}}^2 \wedge b \right) \leq 2Z + \frac{2}{p} (1 + b) \ln \frac{\det V_n}{\det V_0}
$$

$$
\leq \frac{4}{p} (1 + b) \ln \frac{\ln(2bn \vee 2) 5.2 \det V_n}{\delta \det V_0}
$$

which gives the desired statement. $\qquad\square$

**Lemma 26** (Time-uniform Bernstein bound). *In the terminology of [22], let $S_t = \sum_{i=1}^{t} Y_i$ be a sub-$\psi_P$ process with parameter $c > 0$ and variance process $W_t$. Then with probability at least $1 - \delta$ for all $t \in \mathbb{N}$*

$$
S_t \leq 1.44 \sqrt{(W_t \vee m) \left( 1.4 \log \log \left( 2 \left( \frac{W_t}{m} \vee 1 \right) \right) + \log \frac{5.2}{\delta} \right)}
$$

$$
+ 0.41 c \left( 1.4 \log \log \left( 2 \left( \frac{W_t}{m} \vee 1 \right) \right) + \log \frac{5.2}{\delta} \right)
$$

*where $m > 0$ is arbitrary but fixed. This holds in particular when $W_t = \sum_{i=1}^{t} \mathbb{E}_{i-1} Y^2$ and $Y_i \leq c$ for all $i \in \mathbb{N}$.*

*Proof.* The proof follows directly from Theorem 1 with the condition in Table 3 and their stitching boundary in Eq. (10) of [22]. $\qquad\square$

**Lemma 27** (Time-uniform Hoeffding bound). *Let $Y_t$ be a a martingale difference sequence and $G_t, H_t$ two predictable sequences such that $-G_t \leq Y_t \leq H_t$. Then with probability at least $1 - \delta$ for all $t \in \mathbb{N}$*

$$
\sum_{i=1}^{t} Y_i \leq 1.44 \sqrt{(W_t \vee m) \left( 1.4 \log \log \left( 2 \left( \frac{W_t}{m} \vee 1 \right) \right) + \log \frac{5.2}{\delta} \right)}
$$

*where $m > 0$ is arbitrary but fixed and $W_t = \frac{1}{4} \sum_{i=1}^{t} (G_i + H_i)^2$.*

*Proof.* We use the results of [22]. In their terminology, Table 3 in that work shows that $\sum_{i=1}^{t} Y_i$ is a sub-$\psi_N$ process with variance process $W_t$. We can thus apply their Theorem 1 with the stitching boundary in their Eq. (10) with $c = 0$. Setting $\eta = 2$ and $s = 1.4$ gives the desired result. $\qquad\square$