# OpenReview forum: "Neural Active Learning with Performance Guarantees"
_NeurIPS.cc/2021/Conference — NeurIPS 2021 Poster_

### Official Review · Reviewer_iCf1 · 2021-07-08

**Rating:** 7
**Confidence:** 4

**Summary:**

This paper presents algorithms for active learning in the non-parametric regimes, through NTK approximation. The resulting algorithm matches the minimax rates for active learning in certain regimes. It also studies the model selection problem with respect to unknown parameters, which is missed in previous analysis with NTK approximation.

**Limitations And Societal Impact:**

Limitations of the paper are mentioned in Section 5. The results are mainly theoretical; to me, it didn't lead to negative societal impacts in the foreseeable future.

**Main Review:**

This paper converts contextual bandit algorithms into active learning algorithms with statistical learning guarantees. Such an approach looks novel and interesting to me. The paper is well-written and easy to follow. Although I didn't have time to carefully check the proofs, the derived results make sense to me (except for some specific questions as follows).

1. Is there any particular reasons to present the "frozen version" of the algorithm in the main body of the paper? The "frozen version" essentially goes back to a linear bandit algorithm to me. Seems to me that the "trained version" of the algorithm is more interesting. Does the "trained version" requires more assumption or lead to a worse bound?

2. Based on my understanding of the model selection algorithm, e.g., Carrol-type of algorithms, they usually cannot achieve regret smaller than \sqrt{T} even with some known rates (see Thm 6.1 in citation [35]). However, it seems that the guarantee of Theorem 2 surpass that limit (let's say just set \gamma = \alpha). Can you provide some explanations for that?

3. It would be nice to provide some intuitions for the sampling distribution defined in Alg2, and how it helps Alg2 to achieve its guarantees.



**Time Spent Reviewing:**

3

---

> ### Author Response · Authors · 2021-08-09
> **Reply to Reviewer iCf1**
>
> - On presenting the frozen version first.
>
> We agree that the trained version is more interesting. We decided to describe the frozen version first for a couple of reasons:
> (1) The frozen version and its analysis are easier to "digest", especially when combined with the model selection procedure in Algorithm 2. (2) The trained version differs from the frozen one only in some specific places that makes it possible to present it as a "diff" to the frozen version. The trained version yields similar guarantees, but at the cost of a larger network, while assuming the existence of a training procedure (Algorithm 4 in Section A.3) having the same properties as the corresponding procedure in [44].
>
> - On linear bandits vs. active learning.
>
> It is perhaps worth emphasizing that the frozen version (Algorithm 1) is $not$ just linear bandits: one has to deal with the label complexity analysis as well, which is not present in the bandit problem -- please see Section A.1.
>
> - On the model selection algorithm and $\sqrt{T}$ regret.
>
> The improved regret guarantees are made possible by the low-noise assumption (e.g., line 106). Also recall that in active learning we are not just concerned with regret bounds in isolation (like in contextual bandits), but with regret bounds as a function of the requested labels.
>
> - On the sampling distribution in Alg. 2.
>
> The sampling distribution plays base learners with small $d_i$ more often than learners with large $d_i$. Note also that $d_i$ is exactly the (instance-dependent) factor in the cumulative regret and label complexity bounds for base learners that are well-specified. This means that base learners with lower regret are chosen more frequently than  base learners that accumulate regret quicker (and similarly for label complexity). In fact, the sampling distribution is chosen so that the total contribution to the cumulative regret of each base learner is roughly equal. As a consequence, the total cumulative regret of Alg. 2 is at most $M$ (number of base learners) times the regret of each base learner, and the best base learner in particular, which is a key property for achieving the guarantees in Theorem 2. Of course, this only works when the base learners are well-specified but the four tests in Alg 2 ensure that all other learners are eliminated eventually. We will try and add some explainatory text to the main body.

---

> > ### Comment · Reviewer_iCf1 · 2021-08-28
> > **Response to Authors**
> >
> > Thank you for your reply. I read the rebuttal and checked the paper again. I would like to make the following suggestions:
> >
> > 1. I agree that Algorithm 1 is not just a linear bandit due to the label complexity analysis. However, I feel like the frozen version is almost the same as the algorithm presented in [16]. Due to this reason, I believe it's better to present the trained version as the main contribution.
> >
> > 2. Although I roughly get the main idea of Alg 2 and believe it's reasonable, I strongly suggest adding more explanations for Algorithm 2.

---

> > > ### Author Response · Authors · 2021-08-30
> > > **Thanks for your additional suggestions**
> > >
> > > We thank the reviewer for their additional suggestions which we will take into duly consideration in drafting the revised version of this paper.
> > >
> > > The Authors

---

### Official Review · Reviewer_BX5E · 2021-07-08

**Rating:** 6
**Confidence:** 1

**Summary:**

NB: The original submission contains an error. The supplementary material contains the fixed version of the submission.

This paper produces the first NTK based active learning method with provable guarantees.

**Limitations And Societal Impact:**

The concepts and notation used here are very challenging to follow for someone not familiar with NTK. This is not a criticism of the authors or their work, but it makes the paper very challenging to read.

The content of the paper is clearly theoretical, but it would be useful to include at least a few (even very simple) experiments to show that:
1) There are regimes where the active learning method actually outperforms random sampling when using the same NTK estimator in the Algorithms (or even just Algorithm 1, with a known and correct complexity).
2) There are regimes where using these NTK estimators with active learning is better than just training a DNN in a standard way without active learning.

Even a toy example which is built specifically for the method (with explanation for why the method works well on this example) would help readers better understand.

nits:
- L165: parametric should be non-parametric
- Beyond the first page, \mathcal{Y} = {-1, 1}, correct (at least for exposition). Is this why x_{t, a} \in \mathcal{X}^2? For a multiclass problem we would have x_{t, a} \in \mathcal{X}^k right? If so it would be clearer to replace \mathcal{Y} with {-1, 1} throughout, but especially in Algorithm 1, where x_{t, a} \in \mathcal{X}^2 and so the method is already specialized to binary classification.



**Main Review:**

The paper is an extension of the work in Neural Contextual Bandits with UCB-based Exploration (which the authors freely acknowledge), where in moving to an active learning framework the authors had to augment the existing bandit algorithm to account for two key differences:
1) In the bandit algorithm the reward is always seen for free, whereas in active learning the algorithm must decide whether to see the reward.
2) There is a data dependant complexity term which is a required input to the algorithm.

To account for these, the authors first derive a selection algorithm under frozen MLP weights and known complexity. They then deal with the issue of unknown complexity by using a pool of versions of the first algorithm, with various elimination rules for pruning that pool. Additionally in the appendix, they provide versions of both these methods where the MLP weights are not frozen, but are updated by the data as well. For all methods they also provide simultaneous bounds high probability bounds on both the number of examples queried, and the regret

**Time Spent Reviewing:**

2-3

---

> ### Author Response · Authors · 2021-08-09
> **Reply to Reviewer BX5E**
>
> - On the lack of experiments.
>
> Yes, we understand this concern. As shared with the other reviewers, we plan to run some experiments. Yet, as it currently stands, this is a theoretical paper, and we would like it to be viewed as such.
>
> - L. 165.
>
> No, "parametric" is the right term there: $\beta$ determines the complexity of the class and, as $\beta$ grows large, we essentially switch from a non-parametric to a parametric ($\beta \rightarrow \infty$) regime.
>
> - On $\mathcal{Y}$ and binary vs. multiclass, etc.
>
> Yes, due to the reduction to bandits, in the multiclass case, we would have $x_{t,a} \in \mathcal{X}^k$. Thanks for the suggestions on $\mathcal{Y}$ in the pseudocode of the algorithms.

---

### Official Review · Reviewer_DHHc · 2021-07-29

**Rating:** 7
**Confidence:** 1

**Summary:**

Using a combination of the Neural Tangent Kernel (NTK) approximation and contextual bandit approaches, the paper first derives a neural active learning algorithm for the streaming setting where the data points come sequentially and the algorithm has to decide whether to query for the label of each data point. A second similar algorithm is developed for model selection so that the model complexity does not have to be known beforehand. Both algorithms have counterparts which train the neural networks with stochastic gradient descent. The goal of the algorithms is to balance the regret and the number of queried data points. Using the mentioned theoretical frameworks, joint bounds on the regret and the number of queries are derived.

**Limitations And Societal Impact:**

I would say that the main limitation of the paper is the setting to which it is constrained.

**Main Review:**

**Originality**: According to the authors, guarantees on efficient non-parametric active learning algorithms did not exist before their work.

The proposed approach is an adaptation of NeuralUCB [1] to active learning and thus borrows a lot from it. From that point of view, this adaptation is the one of the main contributions of the paper. The other main contribution is the model selection algorithm, while inspired from existing approaches, also seems novel and provides a really nice approximation of the complexity parameter. This latter contribution is essential to the paper as it ties loose ends of the first algorithm.

In relation to related works, the authors noted:

> We emphasize that most of these works, that heavily rely on approximation theory, are not readily comparable to ours, since our goal here is not to approximate h through a DNN on the entire input domain, but only on the data at hand.

However, while I am not knowledgeable of the field, it seems weird to me that comparisons cannot be made even just to highlight the reasons why the proposed approach or the existing approaches cannot be adapted to one setting or the other.

**Quality**: By lack of knowledge of the previous works, I did not verify the proofs and barely understood the main theorems. However, proofs are short, straight to the point and divided into lemmas, which can ease up their understanding.

**Clarity**: The paper is well written for readers already knowledgeable of the three fields it is borrowing from (NTK, contextual bandits and model selection in bandits) in addition to the active learning theory field. If one is not knowledgeable in one of the fields, it might be hard to fully understand the whole. I think adding a paragraph explaining on how all these fields tie together in this paper could improve clarity a lot.

For the order of the sections, I’m not sure why the section Related work goes in Preliminaries and Notation. It should be a section on its own after the introduction.

**Significance**: The results are important in the field of theoretical active learning. It provides theoretical guarantees for using neural networks in active learning with the NTK framework. However, even though the algorithms seem implementable (i.e. not just of theoretical interest), no experiments were done to quantify the practicality of the bounds. One could thus conclude that the bounds are probably vacuous and not of use in practice.

**Ways to improve the paper**: One way to improve would be to do experiments showing how the approach empirically performs. Another way would be to compare the approach to other approaches in similar settings as mentioned in the originality paragraphs.

[1] Zhou, D., Li, L., & Gu, Q. (2020, November). Neural contextual bandits with ucb-based exploration. In International Conference on Machine Learning (pp. 11492-11502). PMLR.


**Time Spent Reviewing:**

20

---

> ### Author Response · Authors · 2021-08-09
> **Reply to Reviewer DHHc**
>
> - On comparison to the relevant literature.
>
> The reviewer is raising a valid point, one with which we have been struggling a lot ourselves. The crux of the matter is that our bounds are data-$dependent$, in that the quantities appearing in the right-hand side are random variables depending on the data at hand (which are themselves random), for instance, so is $S_{T,n}(h)$ in Thm 2. One may attempt to turn these into data $ independent$ results (like in most of the papers we cited in the related work section) by, e.g., establishing bounds on $S_{T,n}(h)$ that hold in expectation or with high probability over the random draw of the data, but this theory is $currently$ $unavailable$ in the NTK literature (as far as we know).
> So, in order to carry out this comparison, we would have to develop ourselves an ad hoc approximation theory associated with NTK. Very recently some results have appeared for certain special cases. See (https://arxiv.org/abs/2102.02336) for example.
> But such results are too embryonic in nature to allow us a full-fledged comparison.
>
> - Improved clarity.
>
> Thanks for the suggestion, we will try and add a more introductory section.
>
> - On improving the paper by adding experiments.
>
> Yes, we agree, this would be a valuable addition. Some experiments are actually planned. As it stands, this is a theoretical paper, and we would like it to be viewed as such.

---

> > ### Comment · Reviewer_DHHc · 2021-09-10
> > **Thank you.**
> >
> > Thank you. Your response was enlightening.

---

### Official Review · Reviewer_FuZA · 2021-07-30

**Rating:** 7
**Confidence:** 3

**Summary:**

This work tackles the problem of active learning for non-parametric classifiers and streaming data, where data points arrive sequentially and, after providing its prediction, the active learning strategy must decide whether to observe the true label or not. Performance of active learning strategies under this setting is evaluated using the difference between the cumulative sum of expected loss incurred by the strategy and the cumulative sum of expected loss incurred by a Bayesian-optimal classifier, which relates to the pseudo-regret in stochastic multi-armed bandits. The authors propose the first computationally efficient algorithm for active learning using Deep Neural Networks (DNNs) (contribution 1), which is achieved using over-parameterized DNNs. By relying on the theory of Neural Tangent Kernel (NTK) approximation, they provide a theoretical analysis of the proposed strategy (contribution 2) and show that one can leverage recent works in model selection to design a novel model selection algorithm (contribution 3). More specifically, they provide high-probability upper-bounds on the pseudo-regret and on the amount of data required for the bound to hold.

**Main Review:**

This paper leverages recent advances in contextual bandit using DNNs with theoretical guarantees for the specific task of active learning under the streaming setting.

[+] The paper is clearly written, well organized, and very easy to follow. Notation and preliminaries are clear and provide all the important information for understanding the paper.

[+] The setting tackled in this work has two components: non-parametric and streaming. It seems novel to provide a method with theoretical guarantees on sample complexity under these two conditions, most of the work in the non-parametric setting focusing on the pool-based situation (rather than streaming).

[-] Alg. 1 is an adaptation of a recent contextual bandit strategy. Application to the active learning streaming setting is natural due to the streaming process. Previous usage of (contextual) bandit methods for active learning are missing from the related work and should be discussed, e.g. Bouneffouf, Laroche, Urvoy, Féraud, and Allesiardo (2014). Contextual bandit for active learning: Active Thompson sampling. In International Conference on Neural Information Processing (pp. 405-412).

[+] By removing the typical assumption (in the contextual bandit setting) that the function complexity is known, Alg. 2 could inspire the development of new methods in the bandit community.

[-] Although the related work section is exhaustive, the second paragraph (lines 153-170) reads like a listing of historical facts. It would benefit from 1-2 sentences highlighting the novelty of the current work compared with the listed works. From what I understand, the current work uses less restrictive assumptions on the marginal distribution -- this should be clarified.

[-] The paper currently lacks experiments to confirm that the proposed method behaves as expected given the theory and that there is indeed an advantage in using it compared with existing approaches. The theoretical arguments seem sound, but it is difficult to assess their relevance and impact without an empirical evaluation. However, I understand that this could be due to the limited space (space is currently used for detailed explanations and discussions of the theoretical results, which is very good).

Disclaimer: I didn't review the proofs in the Appendix.

**Time Spent Reviewing:**

2

---

> ### Author Response · Authors · 2021-08-09
> **Reply to Reviewer FuZA**
>
> - Missing references (e.g., Bouneffouf et al. 2014).
>
> Thanks for pointing this out. We will reference to and add a discussion on that paper. Yet, strictly speaking, that paper $does$ $not$ seem to operate in a selective sampling regime and, besides, is purely experimental in nature.
>
> - Comparison to related work section.
>
> Yes, we have both less assumptions on the marginal distribution and less assumptions on the class of functions $\{h\}$. We will try to re-organize this section so as to better emphasize our contribution -- thanks for the suggestions.
>
> - On the lack of experiments.
>
> Yes, we understand this concern, some experiments are indeed planned. Yet, as it currently stands, this is a theoretical paper, and we would like it to be viewed as such.

---

> > ### Comment · Reviewer_FuZA · 2021-08-31
> > **I have read the rebuttal**
> >
> > Thank you for clarifying these points!

---

### Decision · Program_Chairs · 2021-09-27

**Decision:**

Accept (Poster)

**Comment:**

In the context of streaming data, this paper addresses active learning through a neural contextual bandit and derives theoretical guarantees by relying on the theory of Neural Tangent Kernel (NTK) approximation.The contributions (novel algorithm and its theoretical analysis) are original and relevant. Most of the questions from the reviewers were addressed during the rebuttal. There are still a few concerns about related works and experimental works that the authors are kindly asked to take into account when preparing the final version.